



**Evaluation of global terrestrial evapotranspiration by state-of-the-art**
**approaches in remote sensing, machine learning, and land surface models**
Shufen Pan[1], Naiqing Pan[1,2], Hanqin Tian[1], Pierre Friedlingstein[3], Stephen Sitch[4], Hao Shi[1],
Vivek K. Arora[5], Vanessa Haverd[6], Atul K. Jain[7], Etsushi Kato[8], Sebastian Lienert[9], Danica
Lombardozzi[10], Catherine Ottle[11], Benjamin Poulter[12,13], Sönke Zaehle[14]
[1]International Center for Climate and Global Change Research, School of Forestry and Wildlife
Sciences, Auburn University, Auburn, AL 36832, USA
[2]State Key Laboratory of Urban and Regional Ecology, Research Center for Eco-Environmental
Sciences, Chinese Academy of Sciences, Beijing 100085, China
[3]College of Engineering, Mathematics and Physical Sciences, University of Exeter, Exeter EX4
4QF, United Kingdom
[4]College of Life and Environmental Sciences, University of Exeter, Exeter EX4 4RJ, United
Kingdom
[5]Canadian Centre for Climate Modelling and Analysis, Environment Canada, University of
Victoria, Victoria, BC, Canada
[6]CSIRO Oceans and Atmosphere, GPO Box 1700, Canberra, ACT 2601, Australia
[7]Department of Atmospheric Sciences, University of Illinois, Urbana, IL 61801, USA
[8]Institute of Applied Energy (IAE), Minato-ku, Tokyo 105-0003, Japan
[9]Climate and Environmental Physics, Physics Institute, University of Bern, Bern, Switzerland
[10]Climate and Global Dynamics Laboratory, National Center for Atmospheric Research,
Boulder, CO 80305, USA.
[11]LSCE-IPSL-CNRS, Orme des Merisiers, 91191, Gif-sur-Yvette, France



[12]NASA Goddard Space Flight Center, Biospheric Science Laboratory, Greenbelt, MD 20771,
USA
[13]Department of Ecology, Montana State University, Bozeman, MT 59717, USA
[14]Max Planck Institute for Biogeochemistry, P.O. Box 600164, Hans-Knöll-Str. 10, 07745 Jena,
Germany
**Corresponding author**: panshuf@auburn.edu, Tel: 1-334-844-1015
**Abstract**
Evapotranspiration (ET) is a critical component in global water cycle and links terrestrial water,
carbon and energy cycles. Accurate estimate of terrestrial ET is important for hydrological,
meteorological, and agricultural research and applications, such as quantifying surface energy and
water budgets, weather forecasting, and scheduling of irrigation. However, direct measurement of
global terrestrial ET is not feasible. Here, we first gave a retrospective introduction to the basic
theory and recent developments of state-of-the-art approaches for estimating global terrestrial ET,
including remote sensing-based physical models, machine learning algorithms and land surface
models (LSMs). Then, we utilized six remote sensing-based models (including four physical
models and two machine learning algorithms) and fourteen LSMs to analyze the spatial and
temporal variations in global terrestrial ET. The results showed that the mean annual global
terrestrial ET ranged from $50.7 \times 10^3$ km$^3$ yr$^{-1}$ (454 mm yr$^{-1}$) to $75.7 \times 10^3$ km$^3$ yr$^{-1}$ (697 mm yr$^{-1}$
$^1$), with the average being $65.5 \times 10^3$ km$^3$ yr$^{-1}$(588 mm yr$^{-1}$), during 1982-2011. LSMs had
significant uncertainty in the ET magnitude in tropical regions especially the Amazon Basin, while
remote sensing-based ET products showed larger inter-model range in arid and semi-arid regions
than LSMs.  LSMs and remote sensing-based physical models presented much larger inter-annual



variability (IAV) of ET than machine learning algorithms in southwestern U.S. and the Southern
Hemisphere, particularly in Australia. LSMs suggested stronger control of precipitation on ET
IAV than remote sensing-based models. The ensemble remote sensing-based physical models and
machine-learning algorithm suggested significant increasing trends in global terrestrial ET at the
rate of 0.62 mm $yr^{-2}$ ($p<0.05$) and 0.38 mm $yr^{-2}$ , respectively. In contrast, the ensemble mean of
LSMs showed no statistically significant change (0.23 mm $yr^{-2}$, $p>0.05$), even though most of the
individual LSMs reproduced the increasing trend. Moreover, all models suggested a positive effect
of vegetation greening on ET intensification. Spatially, all methods showed that ET significantly
increased in western and southern Africa, western India and northeastern Australia, but decreased
severely in southwestern U.S., southern South America and Mongolia. Discrepancies in ET trend
mainly appeared in tropical regions like the Amazon Basin. The ensemble means of the three ET
categories showed generally good consistency, however, considerable uncertainties still exist in
both the temporal and spatial variations in global ET estimates. The uncertainties were induced by
multiple factors, including parameterization of land processes, meteorological forcing, lack of in
situ measurements, remote sensing acquisition and scaling effects. Improvements in the
representation of water stress and canopy dynamics are essentially needed to reduce uncertainty in
LSM-simulated ET. Utilization of latest satellite sensors and deep learning methods, theoretical
advancements in nonequilibrium thermodynamics, and application of integrated methods that fuse
different ET estimates or relevant key biophysical variables will improve the accuracy of remote
sensing-based models.
**Keywords**: Evapotranspiration; Land surface models; Remote sensing; Machine learning.





## 1. Introduction

Terrestrial evapotranspiration (ET) is the sum of the water loss to the atmosphere from plant tissues via transpiration and that from the land surface elements including soil, plants and open water bodies through evaporation. Processes controlling ET play a central role in linking the energy (latent heat), water (moisture flux) and carbon cycles (photosynthesis-transpiration trade-off) of the atmosphere, hydrosphere and biosphere. Over 60% of precipitation on the land surface is returned to the atmosphere through ET (Oki and Kanae, 2006), and the accompanying latent heat ($\lambda$ET, $\lambda$ is the latent heat of vaporization) accounts for more than half of the solar energy received by the land surface (Trenberth et al., 2009). ET is also coupled with the carbon dioxide exchange between canopy and atmosphere through vegetation photosynthesis. These linkages make ET an important variable in both the short-term numerical weather predication and long-term climate simulations. Moreover, ET is an excellent indicator for ecosystem functions across a variety of spatial scales. Accurate estimation of land surface ET and understanding of the underlying mechanisms that affect ET variability are therefore essentially required to address a series of climatic, hydrological, ecological and economic issues such as global warming, runoff yield, droughts and agricultural production.

However, there still exists large uncertainty in quantifying the magnitude of global terrestrial ET and its spatial and temporal patterns, despite extensive research (Allen et al., 1998; Liu et al., 2008; Miralles et al., 2016; Mueller et al., 2011; Tian et al., 2010). The previous estimates of global land mean annual ET range from 417 mm year$^{-1}$ to 650 mm year$^{-1}$ for the whole or part of the 1982-2011 period (Mu et al., 2007; Mueller et al., 2011; Vinukollu et al., 2011a; Zhang et al., 2010). This large discrepancy among independent studies may be attributed to lack of sufficient measurements, uncertainty in forcing data, inconsistent spatial and temporal resolutions, ill-



calibrated model parameters and deficiencies in model structures. Of the four components of ET,
transpiration ($T_v$) contributes the largest uncertainty, as it is modulated not only by surface
meteorological conditions and soil moisture but also the physiology and structures of plants.
Changes in non-climatic factors such as elevated atmospheric $CO_2$, nitrogen deposition, and land
covers also serve as influential drivers of $T_v$ (Gedney et al., 2006; Mao et al., 2015; Pan et al.,
2018b; Piao et al., 2010). As such, the global ratio of transpiration to ET ($T_v$/ET) has long been of
debate, with the most recent observation-based estimate being 0.64±0.13 constrained by the global
water-isotope budget (Good et al., 2015). Most earth system models are thought to largely
underestimate $T_v$/ET (Lian et al., 2018).
Global warming is expected to accelerate the hydrological cycle (Pan et al., 2015). For the period,
1982 to the late 1990s, ET was reported to increase by about 7 mm (~1.2%) per decade driven by
rising radiative forcing and temperature (Douville et al., 2013; Jung et al., 2010; Wang et al., 2010).
The contemporary near-surface specific humidity also increased over both land and ocean (Dai,
2006; Simmons et al., 2010; Willett et al., 2007). More recent studies confirm that, since the 1980s,
global ET shows an overall increase (Mao et al., 2015; Yao et al., 2016; Zeng et al., 2018a; Zeng
et al., 2012; Zeng et al., 2016; Zhang et al., 2015; Zhang et al., 2016b). However, the magnitude
and spatial distribution of such a trend are far from determined. Over the past 50 years, pan
evaporation decreased throughout the world (Fu et al., 2009; Peterson et al., 1995; Roderick and
Farquhar, 2002), implying a declining tendency of ET. Moreover, the increase in global terrestrial
ET was found to cease or be even reversed during 1998 to 2008, primarily due to the decreased
soil moisture supply in the Southern Hemisphere (Jung et al., 2010). To reconcile the disparity,
Douville et al. (2013) argued that the peak ET in 1998 should not be taken as a tipping point
because ET was estimated to increase in the multi-decadal evolution. More efforts are needed to





understand the spatial and temporal variations of global terrestrial ET and the underlying
mechanisms that control its magnitude and variability.
Conventional techniques, such as lysimeter, eddy covariance, large aperture scintillometer and the
Bowen ratio method, are capable of providing ET measurements at point and local scales (Wang
and Dickinson, 2012). However, it is difficult to directly measure ET at the global scale because
dense global coverage by such instruments is not feasible and the representativeness of point-scale
measurements to comprehensively represent the spatial heterogeneity of global land surface is also
doubtful (Mueller et al., 2011). To address this issue, numerous approaches have been proposed
in recent years to estimate global terrestrial ET and these approaches can be divided into three
main categories: 1) remote sensing-based physical models, 2) machine learning methods, and 3)
land surface models (Miralles et al., 2011; Mueller et al., 2011; Wang and Dickinson, 2012).
Knowledge of the uncertainties in global terrestrial ET estimates from different approaches is the
prerequisite for future projection and many other applications. In recent years, several studies have
compared multiple terrestrial ET estimates (Khan et al., 2018; Mueller et al., 2013; Wartenburger
et al., 2018; Zhang et al., 2016b). However, most of these studies just analyzed multiple datasets
of the same approach or focused on investigating similarities and differences among different
approaches. Few studies have been conducted to identify uncertainties in multiple estimates of
different approaches.
In this study, we integrate state-of-the-art estimates of global terrestrial ET, including data-driven
and process-based estimates, to assess its spatial pattern, inter-annual variability, climatic drivers,
long-term trend, and reaction to vegetation greening. Our goal is not to compare the various models
and choose the best one, but to identify the uncertainty sources in each type of estimate and provide
suggestions for future model development. In the following sections, we first have a brief



introduction to all methodological approaches and ET datasets used in this study. Second, we
quantify the spatiotemporal variations in global terrestrial ET during the period 1982-2011 by
analyzing the results from the current state-of-the-art models. Finally, we discuss the required
solutions for overcoming the uncertainties identified.
**2. Methodology and data sources**
**2.1 Overview of approaches to global ET estimation**
**2.1.1 Remote sensing-based physical models**
Satellite remote sensing has been widely recognized as a promising tool to estimate global ET,
because it is capable of providing spatially and temporally continuous measurements of critical
biophysical parameters affecting ET, including vegetation states, albedo, fraction of absorbed
photosynthetically active radiation, land surface temperature and plant functional types (Li et al.,
2009). Since the 1980s, a large number of methods have been developed using a variety of satellite
observations (Zhang et al., 2016a). However, part of these methods such as surface energy balance
(SEB) models and surface temperature-vegetation index ($Ts$-VI) space methods are usually applied
at local and regional scales. At the global scales, the vast majority of existing remote sensing-based
physical models can be categorized into two groups: the Penman-Monteith (PM) based and the
Priestley-Taylor (PT) based models.
A) Remote sensing models based on Penman-Monteith equation
The Penman equation, derived from the Monin-Obukhov similarity theory and surface energy
balance, uses surface net radiation, temperature, humidity, wind speed and ground heat flux to
estimate ET from an open water surface. For vegetated surfaces, canopy resistance was introduced
into the Penman equation by Monteith (Monteith, 1965) and the PM equation is formulated as:



$$\lambda \text{ET} = \frac{\Delta(R_n-G)+\rho_a C_p VPD/r_a}{\Delta+\gamma(1+{}^{r_s}/r_a)}$$
(1)

where $\Delta$, $R_n$, G, $\rho_a$, $C_p$, $\gamma$, $r_s$, $r_a$, VPD are the slope of the curve relating saturated water vapor
pressure to air temperature, net radiation, soil heat flux, air density, the specific heat of air,
psychrometric constant, surface resistance, aerodynamic resistance and vapor pressure deficit,
respectively. The canopy resistance term in the PM equation exerts a strong control on
transpiration. For example, based on the algorithm proposed by Cleugh et al. (2007), the MODIS
(Moderate Resolution Imaging Spectroradiometer) ET algorithm improved the model performance
through inclusion of environmental stress into canopy conductance calculation and explicitly
accounted for soil evaporation (Mu et al., 2007). Further, Mu et al. (2011) improved the MODIS
ET algorithm by considering nighttime ET, adding soil heat flux calculation, separating dry canopy
surface from the wet, and dividing soil surface into saturated wet surface and moist surface.
Similarly, Zhang et al. (2010) developed a Jarvis-Stewart-type canopy conductance model based
on normalized difference vegetation index (NDVI) to take advantage of the long-term Advanced
Very High Resolution Radiometer (AVHRR) dataset. More recently, this model was improved by
adding a $CO_2$ constraint function in the canopy conductance estimate (Zhang et al., 2015). Another
important revision for the PM approach is proposed by Leuning et al. (2008). The Penman-
Monteith-Leuning method adopts a simple biophysical model for canopy conductance, which can
account for influences of radiation and atmospheric humidity deficit. Additionally, it introduces a
simpler soil evaporation algorithm than that proposed by Mu et al. (2007), which potentially makes
it attractive to use with remote sensing. However, PM-based models have one intrinsic weakness:
temporal upscaling which is required in translating instantaneous ET estimation into a longer time-
scale value (Li et al., 2009).This could be easily done at the daily scale under clear-sky conditions
but faces challenge at weekly to monthly time-scales due to lack of the cloud coverage information.



B) Remote sensing models based on Priestley-Taylor equation
The Priestley–Taylor (PT) equation is a simplification of the PM equation without parameterizing
aerodynamic and surface conductances (Priestley and Taylor, 1972) and can be expressed as:
$$\lambda \text{ET} = f_{stress} \times \alpha \times \frac{\Delta}{\Delta + \gamma} \times (R_n - G) \qquad (2)$$
where $f_{stress}$ is a stress factor and is usually computed as a function of environmental conditions. $\alpha$
is the PT parameter with a value of 1.2–1.3 under water unstressed conditions and can be estimated
using remote sensing. Although the original PT equation works well in estimating potential ET
across most surfaces, the Priestley-Taylor coefficient, $\alpha$, usually needs adjustment to convert
potential ET to actual ET (Zhang et al., 2016a). Instead, Fisher et al. (2008) developed a modified
PT model that keeps $\alpha$ constant but scales down potential ET by ecophysiological constraints and
soil evaporation partitioning. The accuracy of their model has been validated against eddy
covariance measurements conducted at a wide range of climates and plant functional types (Fisher
et al., 2009; Vinukollu et al., 2011b). Following this idea, Yao et al. (2013) further developed a
modified Priestley-Taylor algorithm that constrains soil evaporation using the Apparent Thermal
Inertia derived index of soil water deficit. Miralles et al. (2011) also proposed a novel PT type
model, Global Land surface Evaporation: the Amsterdam Methodology (GLEAM). GLEAM
combines a soil water module, a canopy interception model and a stress module within the PT
equation. The key distinguishing features of this model are the use of microwave-derived soil
moisture, land surface temperature and vegetation density, and the detailed estimation of rainfall
interception loss. In this way, GLEAM minimizes the dependence on static variables, avoids the
need for parameter tuning, and enables the quality of the evaporation estimates to rely on the
accuracy of the satellite inputs (Miralles et al., 2011). Compared with the PM approach, the PT



based approaches avoid the computational complexities of aerodynamic resistance and the
accompanying error propagation. However, the many simplifications and semi-empirical
parameterization of physical processes in the PT based approaches may lower its accuracy.
**2.1.2 VI-based empirical algorithms and machine learning methods**
The principle of empirical ET algorithms is to link observed ET to its controlling environmental
factors through various statistical regressions or machine learning algorithms of different
complexities. The earliest empirical regression method was proposed by Jackson et al. (1977). At
present, the majority of regression models are based on vegetation indices (Glenn et al., 2010),
such as NDVI and enhanced vegetation index (EVI), because of their simplicity, resilience in the
presence of data gaps, utility under a wide range of conditions and connection with vegetation
transpiration capacity (Maselli et al., 2014; Nagler et al., 2005; Yuan et al., 2010). As an alternative
to statistical regression methods, machine learning algorithms have been gaining increased
attention for ET estimation for their ability to capture the complex nonlinear relationships between
ET and its controlling factors (Dou and Yang, 2018). Many conventional machine learning
algorithms, such as artificial neural networks, random forest, and support vector machine based
algorithms have been applied in various ecosystems (Antonopoulos et al., 2016; Chen et al., 2014;
Feng et al., 2017; Shrestha and Shukla, 2015) and have proved to be more accurate in estimating
ET than simple regression models (Antonopoulos et al., 2016; Chen et al., 2014; Kisi et al., 2015;
Shrestha and Shukla, 2015; Tabari et al., 2013). In up-scaling FLUXNET ET to the global scale,
Jung et al. (2010) used the model tree ensemble method to integrate eddy covariance measurements
of ET with satellite remote sensing and surface meteorological data. In a latest study (Bodesheim
et al., 2018), the random forest approach was used to derive global ET at a half-hourly time-scale.
**2.1.3 Process-based land surface models (LSMs)**


Although satellite-derived ET products have provided quantitative investigations of historical
terrestrial ET dynamics, they can only cover a limited temporal record of about four decades. To
obtain terrestrial ET before 1980s and predict future ET dynamics, LSMs are needed, as they are
able to represent a large number of interactions and feedbacks between physical, biological, and
biogeochemical processes in a prognostic way (Jimenez et al., 2011). ET simulation in LSMs is
regulated by multiple biophysical and physiological properties or processes, including but not
limited to stomatal conductance, leaf area, root water uptake, soil water, runoff and sometimes
nutrient uptake (Famiglietti and Wood, 1991; Huang et al., 2016; Lawrence et al., 2007). Although
almost all current LSMs have these components, different parameterization schemes result in
substantial differences in ET estimation (Wartenburger et al., 2018). Therefore, in recent years,
the multi-model ensemble approach has become popular in improving the accuracy of global
terrestrial ET estimation (Mueller et al., 2011; Wartenburger et al., 2018). Yao et al. (2017) showed
that a simple model averaging method or a Bayesian model averaging method is superior to each
individual model in predicting terrestrial ET.
**2.2 Description of ET datasets**
In this study, we evaluate twenty ET products that are based on remote sensing-based physical
models, machine-learning algorithms, and LSMs to investigate the magnitudes and spatial patterns
of global terrestrial ET over recent decades. Table 1 lists the input data, adopted ET algorithms,
limitations, and references for each product. We use a simple model averaging method when
calculating the mean value of multiple models.
Four physically-based remote sensing datasets, including Process-based Land Surface
Evapotranspiration/Heat Fluxes algorithm (P-LSH), Global Land surface Evaporation: the



Amsterdam Methodology (GLEAM), Moderate Resolution Imaging Spectroradiometer (MODIS)
and PML-CSIRO (Penman-Monteith-Leuning), and two machine-learning datasets, including
Random Forest (RF) and Model Tree Ensemble (MTE), are used in our study. Both machine
learning and physical-based remote sensing datasets were considered as benchmark products.
P-LSH, MODIS and PML-CSIRO quantify ET through PM approaches. P-LSH adopts a modified
PM approach coupling with biome-specific canopy conductance determined from NDVI (Zhang
et al., 2010). The modified P-LSH model used in this study also accounts for the influences of
atmospheric $CO_2$ concentrations and wind speed on canopy stomatal conductance and
aerodynamic conductance (Zhang et al., 2015). MODIS ET model is based on the algorithm
proposed by Cleugh et al. (2007). Mu et al. (2007) improved the model performance through the
inclusion of environmental stress into canopy conductance calculation, and explicitly accounting
for soil evaporation by combing complementary relationship hypothesis with PM equation. The
MODIS ET product (MOD16A3) used in this study was further improved by considering night-
time ET, simplifying vegetation cover fraction calculation, adding soil heat flux item, dividing
saturated wet and moist soil, separating dry and wet canopy, as well as modifying algorithms of
aerodynamic resistance, stomatal conductance, and boundary layer resistance (Mu et al., 2011).
PML-CSIRO adopts Penman-Monteith-Leuning algorithm, which calculates surface conductance
and canopy conductance by a biophysical model instead of classic empirical models. The
maximum stomatal conductance is estimated using the trial-and-error method (Zhang et al., 2016b).
Furthermore, for each grid covered by natural vegetation, the PML-CSIRO model constrains ET
at the annual scale using the Budyko hydrometeorological model proposed by Fu (1981). GLEAM
ET calculation is based on PT equation, which requires less model inputs than PM equation, and
the majority of these inputs can be directly achieved from satellite observations. Its rationale is to



make the most of information about evaporation contained in the satellite-based environmental
and climatic observations (Martens et al., 2017; Miralles et al., 2011). Key variables including air
temperature, land surface temperature, precipitation, soil moisture, vegetation optical depth and
snow-water equivalent are satellite-observed. Moreover, the extensive usage of microwave remote
sensing products in GLEAM ensures the accurate estimation of ET under diverse weather
conditions. Here, we use the GLEAM v3.2 version which has overall better quality than previous
version (Martens et al., 2017).
The MTE approach is based on the Tree Induction Algorithm (TRIAL) and Evolving Trees with
Random Growth (ERROR) algorithm (Jung et al., 2009). The TRIAL grows model trees from the
root node and splits at each node with the criterion of minimizing the sum of squared errors of
multiple regressions in both subdomains. ERROR is used to select the model trees that are
independent from each other and have best performances under Schwarz criterion. Canopy fraction
of absorbed photosynthetic active radiation (fAPAR), temperatures, precipitation, relative
humidity, sunshine hours, and potential radiation are used as explanatory variables to train MTE
(Jung et al., 2011). The rationale of random forest (RF) algorithm is generating a set of independent
regression trees through randomly selecting training samples automatically (Breiman, 2001). Each
regression tree is constructed using samples selected by bootstrap sampling method. After fixing
individual tree in entity, the final result is determined by simple averaging. One merit of RF
algorithm is its capability of handling complicated nonlinear problems and high dimensional data
(Xu et al., 2018). For the RF product used in this study, multiple explanatory variables including
enhanced vegetation index, fAPAR, leaf area index, daytime and nighttime land surface
temperature, incoming radiation, top of atmosphere potential radiation, index of water availability
and relative humidity were used to train regression trees (Bodesheim et al., 2018).

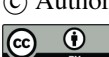



The fourteen LSMs-derived ET products were from the Trends and Drivers of the Regional Scale
Sources and Sinks of Carbon Dioxide (TRENDY) Project (including CABLE, CLASS-CTEM,
CLM45, DLEM, ISAM, JSBACH, JULES, LPJ-GUESS, LPJ-wsl, LPX-Bern, O-CN,
ORCHIDEE, ORCHIDEE-MICT and VISIT). Daily gridded meteorological reanalyses from the
CRU-NCEPv8 dataset (temperature, precipitation, long- and short-wave incoming radiation, wind-
speed, humidity, air pressure) were used to drive the LSMs. The TRENDY simulations were
performed in year 2017 and contributed to the Global Carbon Budget reported in Le Quéré et al.
(2018). We used the results of S3 experiment of TRENDY$_V$6 (with changing $CO_2$, climate and
land use) over the period 1860-2016.
**2.3 Description of other datasets**
To quantify the contributions of vegetation greening to terrestrial ET variations, we used LAI of
TRENDY$_V$6 S3 experiment. We also used the newest version of the Global Inventory Modeling
and Mapping Studies LAI data (GIMMS LAI3gV1) as satellite-derived LAI. GIMMS LAI3gV1
was generated from AVHRR GIMMS NDVI3g using an Artificial Neural Network (ANN) derived
model (Zhu et al., 2013). It covers the period 1982 to 2016 with bimonthly frequency and has a
1/12° spatial resolution. To achieve a uniform resolution, all data were resampled to 1/2° using the
nearest neighbour method. According to Pan et al. (2018a), grids with an annual mean NDVI<0.1
were thought to be non-vegetated regions and were masked. NDVI data were from GIMMS
NDVI3gV1 dataset. Temperature, precipitation and radiation are from CRU-NCEPv8.
**2.4 Statistical analysis**
The significance of ET trends is analyzed using the Mann-Kendall (MK) test (Kendall, 1955; Mann,
1945). It is a rank-based non-parametric method that has been widely applied for detecting a trend





in hydro-climatic time series (Sayemuzzaman and Jha, 2014; Yue et al., 2002). The Theil-Sen
estimator was applied to estimate the magnitude of the slope. The advantage of this method over
ordinary least squares estimator is that it limits the influence of the outliers on the slope (Sen,

318    1968).

Terrestrial ET IAV is mainly controlled by variations in temperature, precipitation, and shortwave
solar radiation (Zeng et al., 2018b; Zhang et al., 2015). In this study, we performed partial
correlation analyses between ET and these three climatic variables at annual scale for each grid
cell to explore climatic controls on ET IAV. Variability caused by climatic variables was assessed
through the square of partial correlation coefficients between ET and temperature, precipitation,
and radiation. We chose partial correlation analysis because it can quantify the linkage between
ET and single environmental driving factor while controlling the effects of other remaining
environmental factors. Partial correlation analysis is a widely applied statistical tool to isolate the
relationship between two variables from the confounding effects of many correlated variables
(Anav et al., 2015; Jung et al., 2017; Peng et al., 2013). All variables were first detrended in the
statistical correlation analysis since we focus on the inter-annual relationship. The study period is
from 1982 to 2011 for all models except MODIS and Rand Forest whose temporal coverage is
limited to 2001-2011 because of data availability.
To quantify the contribution of vegetation greening to terrestrial ET, we separated the trend in
terrestrial ET into four components induced by climatic variables and vegetation dynamics by
establishing a multiple linear regression model between global ET and temperature, precipitation,
shortwave radiation, and LAI (Eq. 3-4):
$$\delta(ET) = \frac{\partial(ET)}{\partial(LAI)}\delta(LAI) + \frac{\partial(ET)}{\partial T}\delta(T) + \frac{\partial(ET)}{\partial(P)}\delta(P) + \frac{\partial(ET)}{\partial R}\delta(R) + \varepsilon \qquad (3)$$



$$\delta(ET) = \gamma_{ET}^{LAI}\delta LAI + \gamma_{ET}^{T}\delta T + \gamma_{ET}^{P}\delta P + +\gamma_{ET}^{R}\delta R + \varepsilon \qquad (4)$$
$\gamma_{ET}^{LAI}$, $\gamma_{ET}^{T}$, $\gamma_{ET}^{P}$, $\gamma_{ET}^{R}$ are the sensitivities of ET to leaf area index (LAI), air temperature (T),
precipitation (P), and radiation (R), respectively. $\varepsilon$ is the residual, representing the impacts of other
factors.
After calculating $\gamma_{ET}^{LAI}$, $\gamma_{ET}^{T}$, $\gamma_{ET}^{P}$, $\gamma_{ET}^{R}$, the contribution of trend in factor i ($Trend(i)$) for the trend
in ET ($Trend(ET)$) can be quantified as follows:
$$Contri(i) = (\gamma_{ET}^{i} \times Trend(i))/Trend(ET) \qquad (5)$$
In performing multiple linear regression, we used GIMMS LAI for both remote sensing-based
physical models and machine learning methods, and used individual TRENDYv6 LAI for each
TRENDY model. Temperature, precipitation and radiation are from CRU-NCEPv8
**3. Results**
**3.1 The ET magnitude estimated by multiple models**

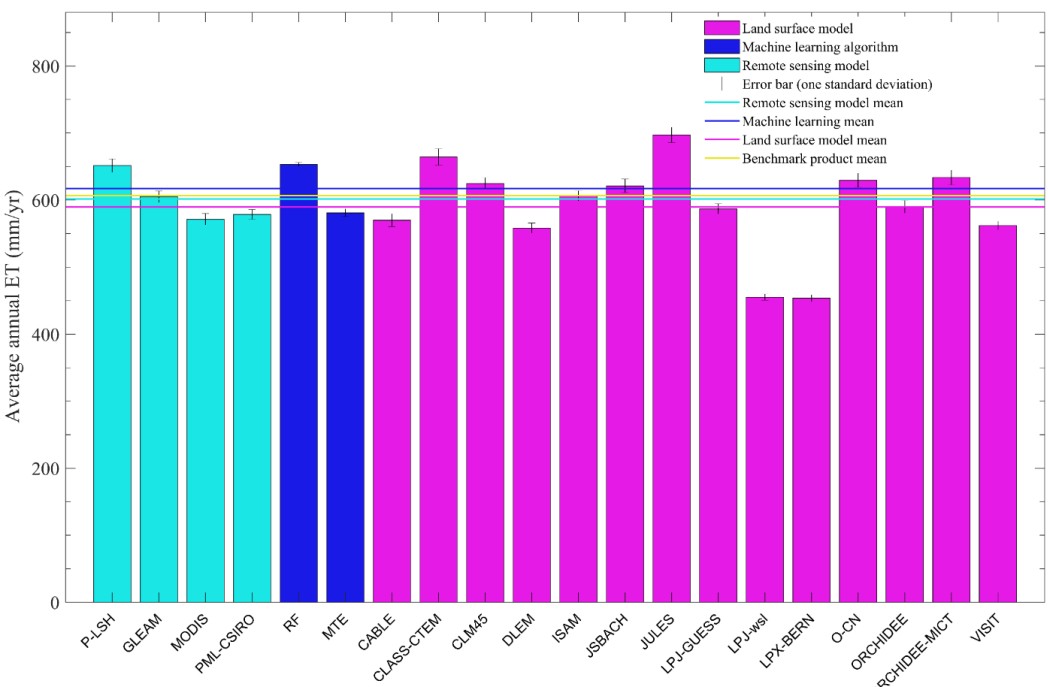


**Figure 1.** Average annual global terrestrial ET estimated by each model during the period 2001-
2011. Error bars represent the standard deviation of each dataset. The four lines indicate the mean
value of each category.

The multi-year ensemble mean of annual global terrestrial ET during 2001-2011 derived by the
machine learning methods, remote sensing methods and TRENDY models agreed well, ranging
from 589.6 mm yr$^{-1}$ to 617.1 mm yr$^{-1}$. However, substantial differences existed among individual
datasets (Fig. 1). LPJ-wsl (455.3 mm yr$^{-1}$) and LPX-Bern (453.7 mm yr$^{-1}$) estimated significantly
lower ET than other models, even in comparison with most previous studies focusing on earlier
periods (Table S1). In contrary, JULES gave the largest ET estimate (697.3 mm yr$^{-1}$, equals to
$7.57 \times 10^4$ km$^3$ yr$^{-1}$) among models used in this study, and showed an obvious increase of ET
compared to its estimation during 1950-2000 ($6.5 \times 10^4$ km$^3$ yr$^{-1}$, Table S1).



### 3.2 Spatial patterns of global terrestrial ET

As shown in Fig. 2, the spatial patterns of multi-year average annual ET derived by different approaches were similar. ET was the highest in tropics and low in northern high latitudes and arid regions such as Australia, central Asia, western US and Sahel. Compared to remote sensing-based physical models and LSMs, machine-learning methods obtained a smaller spatial gradient. In general, latitudinal profiles of ET estimated by different approaches were also consistent (Fig. 3). However, machine-learning methods gave higher ET estimate at high latitudes and lower ET in tropics compared to other approaches. In tropics, LSMs have significant larger uncertainties than benchmark products, and the standard deviation of LSMs is about two times as large as that of benchmark products (Fig. 3). In other latitudes, LSMs and benchmark ET products have generally comparable uncertainties. The largest difference in ET of different categories was found in the Amazon Basin (Fig. 2). In most regions of Amazon Basin, the mean ET of remote sensing physical models are more than 200mm higher than the mean ET of LSMs and machine-learning methods. For individual ET estimate, the largest uncertainty was also found in the Amazon Basin. MODIS, VISIT and CLASS-CTEM estimated that annual ET was larger than 1300 mm in the majority of Amazon, whereas JSBACH and LPJ-wsl estimated ET of smaller than 800 mmyr$^{-1}$ (Fig. S1). As is shown in Fig. S2, the differences in ET estimate among TRENDY models were larger than those among benchmark estimates in tropical and humid regions. The uncertainty of ET estimates by LSMs is particularly large in the Amazon Basin where the standard deviation of LSMs estimates is more than two times as large as that of benchmark estimates. It is noteworthy that, in arid and semi-arid regions such as western Australia, central Asia, northern China and western US, the differences in ET estimate among LSMs is significantly smaller than those among remote sensing models and machine learning algorithms.



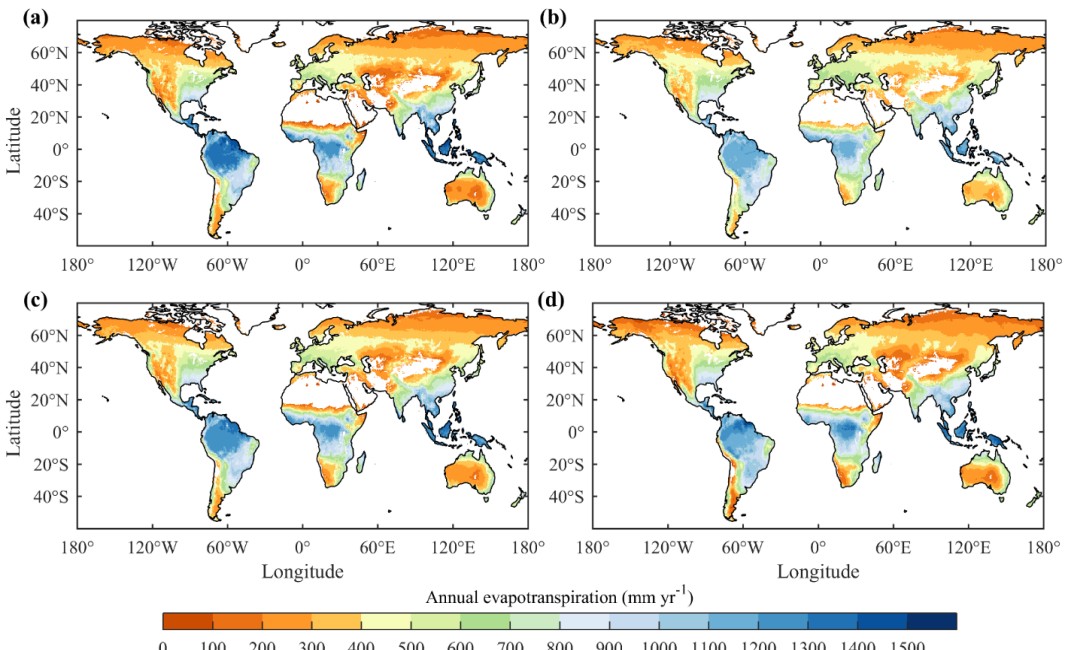


**Figure 2.** Spatial distributions of mean annual ET derived from (a) remote sensing-based physical
models, (b) machine-learning algorithms, (c) benchmark datasets and (d) TRENDY LSMs
ensemble mean, respectively.

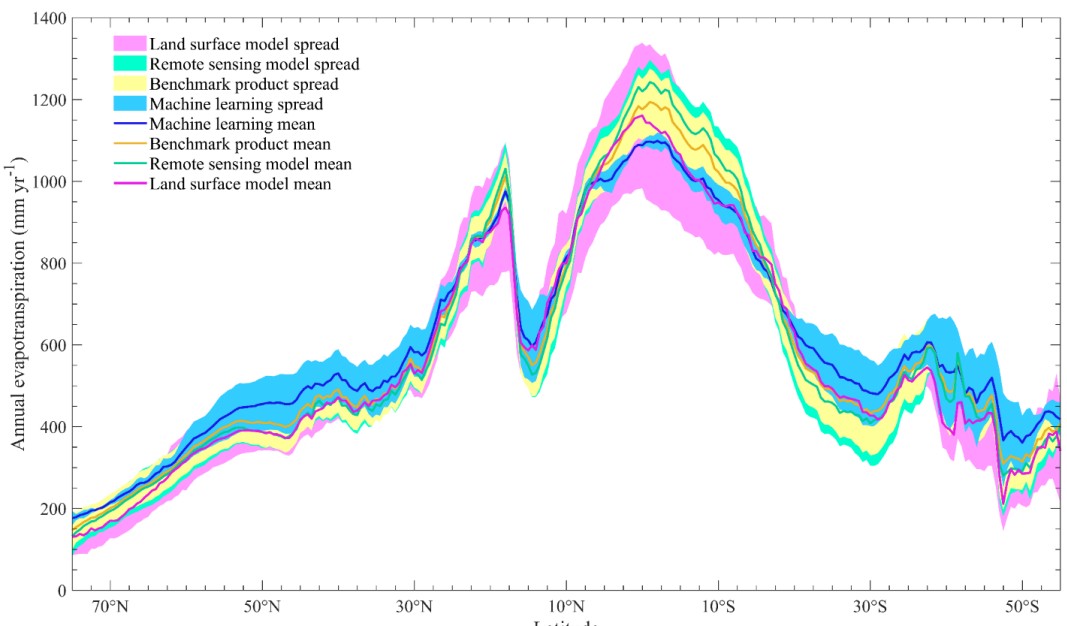


**Figure 3.** Latitudinal profiles of mean annual ET for different categories of models. Each line

represents the mean value of the corresponding category and the shading represents the interval of

one standard deviation.

**3.3 Inter-annual variations in global terrestrial ET**

The ensemble mean inter-annual variability (IAV) of remote sensing ET estimates and LSMs ET

estimates showed similar spatial patterns (Fig. 4). Both remote sensing physical models and LSMs

presented low IAV in ET in northern high latitudes but high IAV in ET in southwestern U.S, India,

south Sahara Africa, Amazon and Australia. In contrast, IAV of machine-learning based ET was

much weaker. In most regions, IAV of machine learning ET smaller than 40% of IAV of remote

sensing physical ET and LSMs ET, and this phenomenon is especially pronounced in tropical

regions. Further investigation into the spatial patterns of ET IAV for individual model showed that

the two machine-learning methods performed equally in estimating spatial patterns of ET IAV





(Fig. S4). In contrast, ET IAV among remote sensing physical estimates and LSMs estimates were

much larger. LSMs showed the largest differences in IAV of ET in tropical regions. For example,

CABLE and JULES obtained an ET IAV of smaller than 15 mm yr$^{-1}$ in most regions of the Amazon

Basin, while LPJ-GUESS predicted an ET IAV of larger than 60 mm yr$^{-1}$. Figure 5 showed that,

in the north of 20ºS, remote sensing physical ET and LSMs ET had comparable IAV, but IAV of

the machine learning based ET was much smaller. In the region south of 20ºS, TRENDY ET

showed the largest IAV, followed by those of remote sensing physical ET and machine learning

estimates. The three categories of models agreed on that ET IAV in the Southern Hemisphere was

generally larger than that in the Northern Hemisphere.

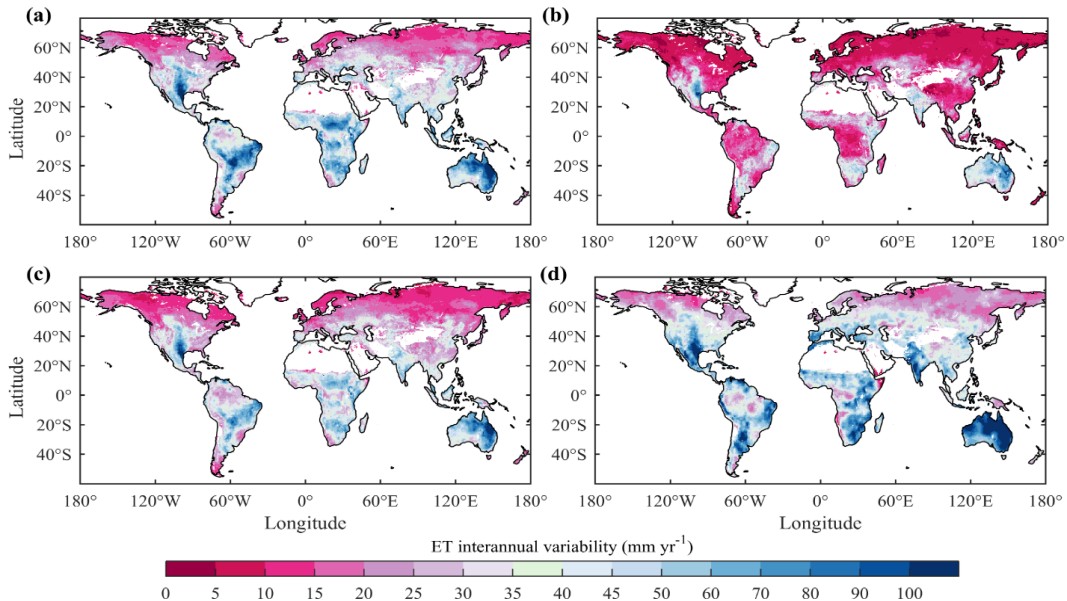

**Figure 4.** Spatial distributions of the inter-annual variability in ET derived from (a) remote

sensing-based physical models, (b) machine learning algorithms, (c) benchmark datasets, and (d)

TRENDY LSMs ensemble mean, respectively. The study used for inter-annual variability analysis

is from 1982 to 2011.

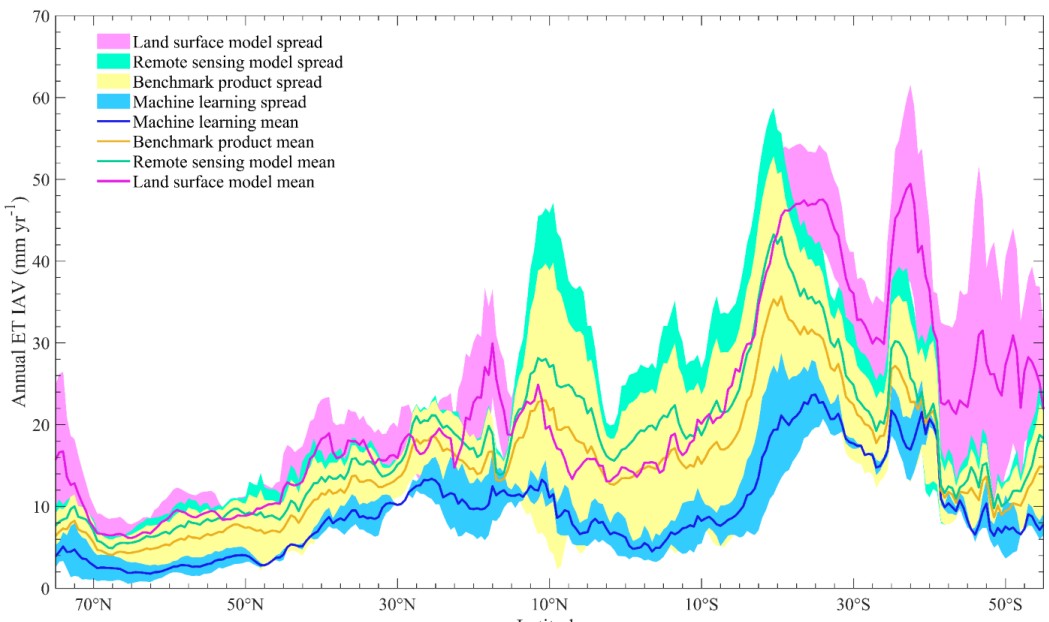


**Figure 5.** Latitudinal profiles of ET IAV for different categories of models. Each line represents the mean value of the corresponding category and the shading represents the interval of one standard deviation.

**3.4 Climatic controls on ET**
According to the ensemble remote sensing models, temperature and radiation dominated ET IAV
in the northern Eurasia, northern and eastern North America, southern China, Congo River Basin
and southern Amazon River Basin, while precipitation dominated ET IAV in arid regions and
semi-arid regions (Fig. 6a). The ensemble machine-learning algorithms had a similar pattern, but
suggested a stronger control of radiation in the Amazon Basin and a weaker control of precipitation
in several arid regions such as central Asia and northern Australia (Fig. 6b). In comparison, the
ensemble LSMs suggested the strongest control of precipitation on ET IAV (Fig. 6). According to
the ensemble LSMs, ET IAV was dominated by precipitation IAV in most regions of the Southern



Hemisphere and northern low latitudes. Temperature and radiation only controlled northern
Eurasia, eastern Canada and part of the Amazon Basin (Fig. 6d). As is shown in Fig. S6, the
majority of LSMs agreed on the dominant role of precipitation in controlling ET in regions south
of 40°N. However, the pattern of climatic controls in the ORCHIDEE-MICT model is quite unique
and different from all other LSMs. According to the ORCHIDEE-MICT model, radiation and
temperature dominate ET IAVs in more regions, and precipitation only controls ET IAVs in
eastern Brazil, northern Russia, central Europe and a part of tropical Africa. Since ORCHIDEE-
MICT was developed from ORCHIDEE, the dynamic root parameterization in ORCHIDEE-MICT
may explain why ET is less driven by Precipitation compared to ORCHIDEE (Haverd et al., 2018).
It is noted that MTE and RF had significant discrepancies in the spatial pattern of dominant
climatic factors. According to the result of MTE, temperature controlled ET IAV in regions north
of 45°N, eastern US, southern China and the Amazon basin (Fig. S6e). By contrast, RF suggested
that precipitation and radiation dominated ET IAV in these regions (Fig. S6f).

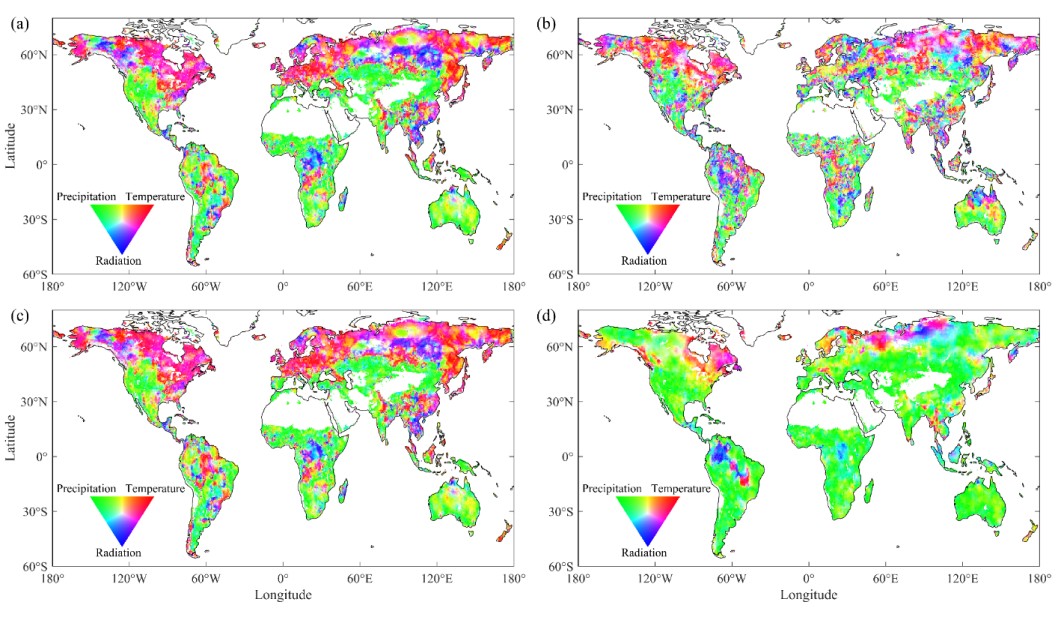




**Figure 6.** Spatial distributions of climatic controls on inter-annual variation of ET derived from
the ensemble means of remote sensing-based physical models (a), machine learning algorithms
(b), benchmark data (c), and TRENDY LSMs (d). (red: temperature; green: precipitation; and blue:
radiation).
**3.5 Long-term trends in global terrestrial ET**
All approaches suggested an overall increasing trend in global ET during the period 1982-2011
(Fig. 7), although ET decreased over 1998-2009. This result is consistent with previous studies
(Jung et al., 2010; Lian et al., 2018; Zhang et al., 2015). Remote sensing physical models indicated
the largest increase in ET (0.62 mm $yr^{-2}$), followed by the machine-learning method (0.38 mm $yr^{-2}$)
$^{-2}$), and land surface models (0.23 mm $yr^{-2}$). Mean ET of all categories except TRENDY models
significantly increased during the study period ($p<0.05$). It is noted that the ensemble mean ET of
different categories are statistically correlated with each other ($p<0.001$), even the driving forces
of different ET approaches are different.


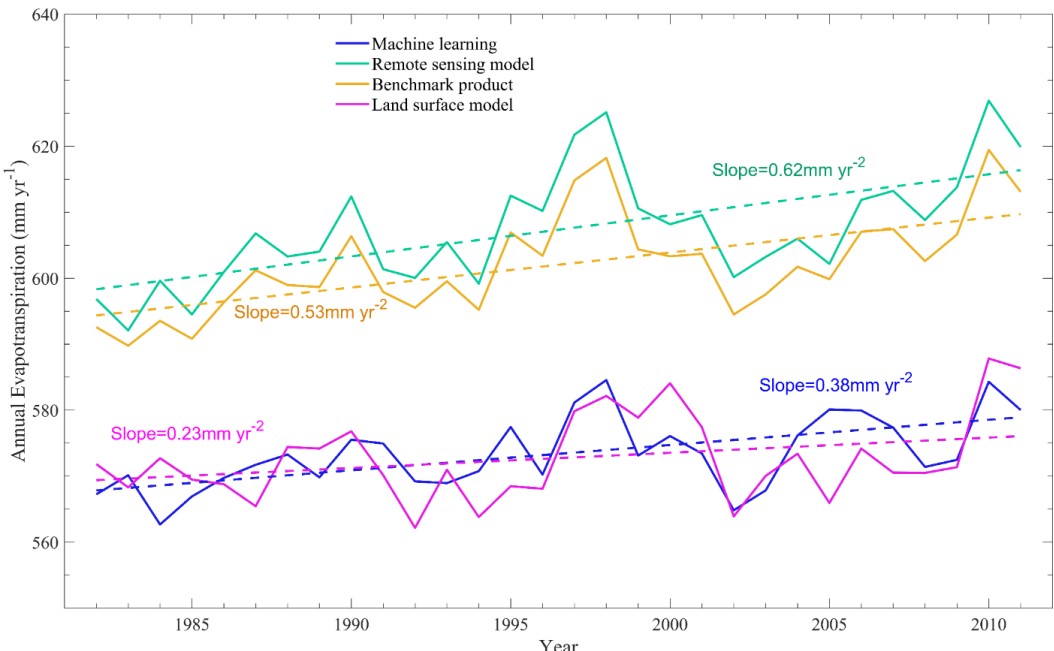


**Figure 7.** Inter-annual variations in global terrestrial ET estimated by different categories of approaches.

All remote sensing and machine learning estimates indicate a significant increasing trend in ET during the study period ($p<0.05$), although the increase rate of P-LSH (1.07 mm yr$^{-2}$) is more than three times as large as that of GLEAM (0.33 mm yr$^{-2}$). Nevertheless, there is a larger discrepancy among LSMs in terms of ET trend. The majority of LSMs (10 of 14) suggest an increasing trend with the average trend of 0.34 mm yr$^{-2}$ ($p<0.05$), and eight of them are statistically significant (see Table 2). However, four LSMs (JSBACH, JULES, ORCHIDEE and ORCHIDEE-MICT) suggest a decreasing trend with the average trend of -0.12 mm yr$^{-2}$ ($p>0.05$) and the trend of ORCHIDEE-MICT (-0.34 mm yr$^{-2}$) is statistically significant ($p<0.05$).





According to Fig. 8, the ensemble means of all the three categories of approaches showed
increasing trends of ET over western and southern Africa, western Indian, and northern Australia,
and decreasing ET over western United States, southern South America and Mongolia.
Discrepancies in ET trends mainly appeared in East Europe, eastern India and central China. LSMs
also suggested larger area of decreasing ET in both North America and South America. Although
the differences in ET trends among individual modes were larger, the majority of models agreed
on that ET increased in western and southern Africa, and decreased in western United States and
southern South America (Fig. S2). For both remote sensing estimates and LSMs estimates, ET
trends in Amazon Basin had large uncertainty. P-LSH, CLM-45 and VISIT suggested large area
of increasing ET, in contrast, GLEAM, JSBACH and ORCHIDEE suggested large area of
decreasing ET.

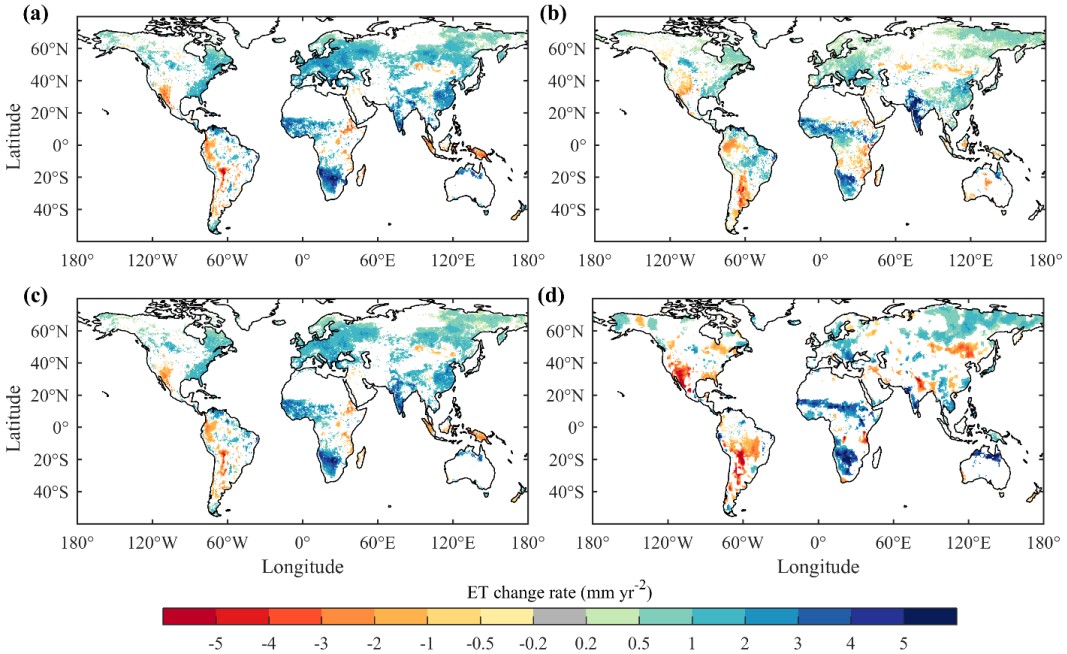






**Figure 8.** Spatial distributions of ET trends during the period 1982-2011 derived from (a) remote
sensing-based physical models, (b) machine learning algorithm, (c) benchmark datasets, and (d)
TRENDY LSMs ensemble mean, respectively. Regions with non-significant trends were excluded.
**3.6 Impacts of vegetation changes on ET variations**
During the period 1982-2011, global LAI trends estimated from remote sensing data and from the
ensemble LSMs are $2.51\times10^{-3}$ $m^2 m^{-2}$ $yr^{-1}$ (p<0.01) and $4.63\times10^{-3}$ $m^2 m^{-2}$ $yr^{-1}$ (p<0.01), respectively
(Table 2). Each LSM suggested a significant increasing trend in global LAI (greening). It was
found that, for both benchmark estimates and LSMs estimates, the spatial pattern of trends in ET
matched well with that of trends in LAI (Fig. 5c-d and Fig. S5a-b), indicating significant effects
of vegetation dynamics on ET variations. According to the results of multiple linear regression, all
models agreed on that greening of the Earth since the early 1980s intensified terrestrial ET (Table
2), although there was a significant discrepancy in the magnitude of ET intensification which
varied from 0.04 $mm$ $yr^{-2}$ to 0.70 mm $yr^{-2}$. The ensemble LSMs suggested a smaller ET increase
(0.23 mm $yr^{-2}$) than the ensemble remote sensing physical models (0.62 mm $yr^{-2}$) and machine-
learning algorithm (0.38 mm $yr^{-2}$). Nevertheless, the greening-induced ET intensification
estimated by LSMs (0.37 mm $yr^{-2}$) is larger than that estimated by remote sensing models (0.28
mm $yr^{-2}$) and machine-learning algorithm (0.09 mm $yr^{-2}$) because LSMs suggested a stronger
greening trend than remote sensing models. The contribution of vegetation greening to ET
intensification estimated by the ensemble LSMs is larger than 100% while that contributions
estimated by the ensemble remote sensing physical models (0.62 mm $yr^{-2}$) and machine-learning
algorithm are smaller than 50%. Although TRENDY LSMs were driven by the same climate data
and remote sensing physical models were driven by varied climate data, TRENDY LSMs still
showed a larger discrepancy in terms of the effect of vegetation greening on terrestrial ET than





remote sensing physical models because of the significant differences in both LAI trend (1.74-
13.63×10⁻³ m² m⁻² yr⁻¹) and the sensitivity of ET to LAI (4.04-217.39 mm yr⁻² per m² m⁻²).
**4.  Discussion and perspectives**
**4.1 Sources of uncertainty**
**4.1.1 Uncertainty in the ET estimation of Amazon Basin**
LSMs have large discrepancies in the magnitude and trend of ET in the Amazon Basin (Fig. 3 and
Fig. S3). However, identifying the uncertainty source is complex. Given that the TRENDY LSMs
used uniform meteorological inputs, the differences of the participating models mainly arise from
the differences in underlying model structures and parameters. One potential source of uncertainty
is the parameterization of root water uptake. In the Amazon Basin, large root depth was confirmed
by field measurements (Nepstad et al., 2004). However, many LSMs have an unrealistically small
rooting depth (generally less than 2 m), neglecting the existence and significance of deep roots.
The incorrect root distributions enlarge the differences in plant available water and root water
uptake, producing large uncertainties in ET. In addition, differences in the parameterization of
other key processes pertinent to ET such as LAI dynamics (Fig. S5), canopy conductance
variations (Table 1), water movements in soil (Abramopoulos et al., 1988; Clark et al., 2015;
Noilhan and Mahfouf, 1996) and soil moisture's control on transpiration (Purdy et al., 2018; Szutu
and Papuga, 2019) also increase the uncertainty in ET. The above-mentioned processes are not
independent of each other but interact in complex ways to produce the end result.
**4.1.2 Uncertainty in the ET estimation of arid and semi-arid regions**
In arid and semi-arid regions, benchmark products show much larger differences in the magnitude
of ET than LSMs (Fig. S2). One cause of this phenomenon is the differences in meteorological





forcing. Remote sensing and machine learning datasets used different forcing data. For
precipitation, RF used CRUNCEPv6 dataset; MTE used Global Precipitation Climatology Centre
(GPCC) dataset; MODIS used Global Modeling and Assimilation Office (GMAO) dataset;
GLEAM used Multi-Source Weighted-Ensemble Precipitation (MSWEP) dataset; PML-CSIRO
used the Princeton Global Forcing (PGF) and the WATCH Forcing Data ERA-Interim (WFDEI)
datasets; and P-LSH used data derived from four independent sources. Since precipitation is the
key climatic factor controlling ET in arid and semi-arid regions (Fig. 6), discrepancies between
different forcing precipitation (Sun et al., 2018) may be the main source of large uncertainty there.
In comparison, the uniform forcing data reduced the inter-model range in ET estimates of
TRENDY LSMs. Nevertheless, it is noted that the congruence across LSMs ET estimates doesn't
necessarily mean they are the correct representation of ET. The narrower inter-model range may
suggest shared biases. All remote sensing models and machine learning algorithms except
GLEAM do not explicitly take the effects of soil moisture into account (Table S1). Given that soil
moisture is pivotal to both canopy conductance and soil evaporation in arid and semi-arid regions
(A et al., 2019; De Kauwe et al., 2015; Medlyn et al., 2015; Purdy et al., 2018), the lack of soil
moisture information also increases the bias in ET estimation. In addition, the accuracy of
remotely-sensing data itself is also an uncertainty source. The retrieval of key land surface
variables, such as leaf area index and surface temperature, is influenced by vegetation architecture,
solar zenith angle and satellite observational angle, particularly over heterogeneous surface
(Norman and Becker, 1995).
**4.1.3 Uncertainty in the ET IAV in the Southern Hemisphere**
In regions south of 20ºS (including Australia, southern Africa and southern South America), the
ET IAVs of remote sensing models and machine learning algorithms are smaller than that of LSMs



(Fig. 4 and 5), although their spatial patterns are similar. In these regions, GLEAM, the only remote
sensing model explicitly considers the effects of soil moisture, has larger ET IAVs than other
remote sensing models and has similar ET IAVs with LSMs (Fig. S4). It implies that most existing
remote sensing models may underestimate ET IAVs in the Southern Hemisphere because the
effects of soil moisture is not explicitly considered. Machine learning algorithms have much
smaller IAVs than other models (Fig. 4 and S4). The main reason is that ET inter-annual variability
is partly neglected in the training process because the magnitude of ET inter-annual variability is
usually smaller than the spatial and seasonal variability (Anav et al., 2015; Jung et al., 2019).
Moreover, the IAV of satellite-based key land surface variables such as LAI, fAPAR and surface
temperature may be not reliable because of the effects of clouds, which also affects the estimation
of IAV of satellite-based ET. It is noted that LSMs ET IAVs show large differences in latitudes
south of 20ºS (Fig. 5). This divergence in ET IAV indicates that land surface models need better
representation of ET response to climate in the Southern Hemisphere.
**4.1.4 Uncertainty in global ET trend**
All of the three categories of ET models detected an overall increasing trend in global terrestrial
ET since the early 1980s, which is in agreement with previous studies (Mao et al., 2015; Miralles
et al., 2014; Zeng et al., 2018a; Zeng et al., 2018b; Zeng et al., 2014; Zhang et al., 2015; Zhang et
al., 2016b). Benchmark products generally suggested stronger ET intensification than LSMs. The
weaker ET intensification in LSMs may be induced by the response of stomatal conductance to
increasing atmospheric $CO_2$ concentration. The increasing $CO_2$ affects ET in two ways. On one
hand, increasing $CO_2$ can effectively reduce stomatal conductance and thus decrease transpiration
(Heijmans et al., 2001; Leipprand and Gerten, 2006; Swann et al., 2016); on the other hand, it can
increase vegetation productivity and thus increase LAI. For benchmarks, the second effect could





be captured by remote sensed LAI, NDVI or fAPAR, while the first effect was neglected by all
models except P-LSH (Zhang et al., 2015). In contrast, both effects were modeled in all TRENDY
LSMs.
LAI dynamics have significant influences on ET. The increased LAI trend (greening) since the
early 1980s was reported by previous studies (Mao et al., 2016; Zhu et al., 2016) and is also
confirmed by remote sensing data and all TRENDY LSMs used in this study (Table 2 and Fig. S5).
Zhang et al. (2015) found that the increasing trend of global terrestrial ET over 1982-2013 was
mainly driven by increase in LAI and the enhanced atmosphere water demand. Using a land–
atmosphere coupled global climate model (GCM), Zeng et al. (2018b) further estimated that global
LAI increased about 8%, resulting in an increase of $0.40\pm0.08$ mm yr$^{-1}$ in global ET (contributing
to 55%±25% of the ET increase). This number is close to the estimates of ensemble LSMs
($0.37\pm0.18$ mm yr$^{-1}$). In comparison, remote sensing models and machine learning algorithm used
in this study suggested smaller greening-induced ET increases. It is noted that TRENDY LSMs
still showed a larger discrepancy in terms of the effect of vegetation greening on terrestrial ET
than remote sensing physical models (Table 2) because of the significant differences in LAI trend
($1.74$-$13.63\times10^{-3}$ m$^2$ m$^{-2}$ yr$^{-1}$) and in the sensitivity of ET to LAI ($4.04$-$217.39$ mm yr$^{-2}$ per m$^2$ m$^{-2}$
$^2$). Uncertainties in LAI trend may arise from inappropriate carbon allocations and deficits in
responding to water deficits (Anav et al., 2013; Hu et al., 2018; Murray-Tortarolo et al., 2013;
Restrepo‐Coupe et al., 2017). Additionally, for machine-learning algorithms, the results from
insufficient long-term in situ measurements and sparse observations in tropical, boreal and arid
regions imply that there likely are deficiencies in representing the temporal variations.
**4.1.5 Ignorance of the effects of irrigation**



Irrigation accounts for about 90% of human consumptive water use and largely effects on ET in
irrigated croplands (Siebert et al., 2010).  Global withdrawal of irrigation was estimated to within
the range of 1161-3800 $km^3yr^{-1}$ around the year 2000, and largely increased during the period
2000-2014 (Chen et al., 2019). However, none of the remote sensing physical models and
machine-learning algorithms explicitly accounted for the effects of irrigation on ET, although these
effects could be taken into account to some extent by using observed LAI, NDVI, or fAPAR to
drive the models (Zhang et al., 2015). Considering that annual ET may surpass annual precipitation
in cropland, Zhang et al. (2016b) used the Budyko hydrometeorological model to constrain PML-
CSIRO model only in grids covered by non-crop vegetation. But the process of irrigation affecting
evaporation was still not taken into consideration. For TRENDY LSMs, only 2 of 14 models
(DLEM and ISAM) included the irrigation processes (Le Quéré et al., 2018). Therefore, the effects
of irrigation are largely neglected in existing global ET datasets, which reduces the accuracy of
local ET estimates in regions with a large proportion of irrigated cropland.
In short, the multi-model inter-comparison indicates that considerable uncertainty exists in both
the temporal and spatial variations in global ET estimates, even though a large portion of models
adopt similar ET algorithms (Table 1). The major uncertainty source could be different for
different types of models and regions. The uncertainty is induced by multiple factors, including
problems pertinent to parameterization of land processes, lack of in situ measurements, remote
sensing acquisition, scaling effects and meteorological forcing.
**4.2 Recommendations for future development**
**4.2.1 Remote sensing-based physical methods**



In the past decades, the development of remote sensing technologies has contributed to the boom
of various ET estimating methods. However, there is still a large room for remote sensing
technologies to improve (Fisher et al., 2017). Developing new platforms and sensors that have
improved global spatiotemporal coverage and using multi-band, multi-source remote sensing data
are the key points. Planned or newly launched satellites, such as NASA's GRACE Follow-On
(GRACE-FO) mission and ECOsystem Spaceborne Thermal Radiometer Experiment on Space
Station (ECOSTRESS) mission, will improve the accuracy of terrestrial ET estimates.
ECOSTRESS's thermal infrared (TIR) multispectral scanner is capable of monitoring diurnal
temperature patterns at high-resolutions, which gives insights into plant response to water stress
and the means to understand sub-daily ET dynamics (Hulley et al., 2017). GRACE Follow-On
observations can be used to constrain subsurface lateral water transfers, which helps to correct soil
moisture and subsequently improves the accuracy of ET estimates (Rouholahnejad and Martens,
2018). Moreover, building integrated methods that fuse different ET estimates or the upstream
satellite-based biophysical variables from different platforms and the other forcing data will be
helpful to improve the accuracy and spatiotemporal coverage of ET (Ke et al., 2016; Ma et al.,
2018; Semmens et al., 2016).
The theories and retrieval algorithms of ET and related key biophysical variables also need to be
further improved. For example, the method for canopy conductance calculation may be improved
by integrating remote sensing based solar-induced chlorophyll fluorescence (SIF) data. SIF data
in existing Global Ozone Monitoring Experiment-2 (GOME-2), Orbiting Carbon Observatory-2
(OCO-2) and TROPOspheric Monitoring Instrument (TROPOMI) and the forthcoming OCO-3
and Geostationary Carbon Cycle Observatory (GeoCarb) satellites provide a good opportunity for
diagnosing transpiration and for ET partitioning at multiple spatiotemporal scales (Pagán et al.,





2019; Stoy et al., 2019; Sun et al., 2017). Theoretical advancements in nonequilibrium
thermodynamics and Maximum Entropy Production (MEP) could be incorporated into the
classical ET theories (Xu et al., 2019; Zhang et al., 2016a). In addition, quantifying the effects of
$CO_2$ fertilization on stomatal conductance is pivotal for remote sensing models to capture the long-
term trend of terrestrial ET.
**4.2.2 Machine learning methods**
It is well known that the capability of machine-learning algorithms in providing accurate ET
estimates largely depends on the representativeness of training datasets in describing ecosystem
behaviors (Yao et al., 2017). As a result, machine-learning algorithms may not perform well
outside the range of the data used for their training. Unfortunately, long-term field observations
out of northern temperate regions are still insufficient; this is an importance cause for the small
spatial gradient and small IAVs of machine-learning ET. Given that remote sensing is capable of
providing broad coverage of key biophysical variables at reasonable spatial and temporal
resolutions, one way to overcome this challenge is to exclusively use remote sensing observations
as training data (Jung et al., 2019; Poon and Kinoshita, 2018). Another simple way to make IAVs
of machine-learning ET more realistic is normalizing the yearly anomalies when comparing with
ET estimates from LSMs and remote sensing physical models (Jung et al., 2019). New machine-
learning techniques, including the extreme learning machine and the adaptive neuro-fuzzy
inference system, can be used to improve the accuracy of ET estimation (Gocic et al., 2016; Kişi
and Tombul, 2013). The emerging deep learning methods such as recurrent neural network (RNN)
and Long Short-Term Memory (LSTM) have large potential to outcompete conventional machine-
learning methods in modelling ET time series (Reichstein et al., 2018; Reichstein et al., 2019).
Almost all machine-learning datasets used precipitation rather soil moisture as explanatory



variable when training. However, soil moisture rather than precipitation directly controls ET. As
more and more global remote sensing based soil moisture datasets become available, using soil
moisture products as input is expected to improve the accuracy of ET estimates, especially for
regions with spares vegetation coverage (Xu et al., 2018).

**4.2.3 Land surface models**


In contrast to observation-based methods, LSMs are able to predict future changes in ET, and can
disentangle the effects of different drivers on ET through factorial analysis. However, results from
LSMs are only as good as their parameterizations of complex land surface processes which are
limited by our incomplete understanding of physical and biological processes (Niu et al., 2011).
Although TRENDY LSMs are the state of the art process-based land surfaces models,
improvements are still needed because several important processes are missing or not being
appropriately parameterized. Most of the TRENDY LSMs did not simulate the processes relevant
to human management including irrigation (Chen et al., 2019) and fertilization (Mao et al., 2015),
and natural disturbances like wildfire (Poon and Kinoshita, 2018). Incorporating these processes
into present LSMs is critical. However, we need to keep it in mind that these processes should be
added with caution, because adding more processes and introducing new model parameters may
lead to an increase in model's uncertainty.
In light of the importance of soil water availability in constraining canopy conductance and
dynamics, accurate representation of hydrological processes is a core task for LSMs, particularly
in dry regions. Integrating a dynamic root water uptake function and hydraulic redistribution into
the LSM can significantly improve its performance of estimating seasonal ET and soil moisture
(Li et al., 2012). Moreover, other hydrological processes including groundwater(Decker, 2015),
lateral flow (Rouholahnejad and Martens, 2018) and water vapor diffusion at the soil surface





(Chang et al., 2018) need to be simulated and correctly represented to reproduce the dynamics of
soil water and ET. Since canopy LAI plays an important role in regulating ET, correctly simulating
vegetation dynamics is also critical. One way is to correct the initialization, distribution, and
parameterization of vegetation phenology in LSMs (Murray-Tortarolo et al., 2013; Zhang et al.,
2019). Appropriate carbon allocation scheme and parameterization of vegetation's response to
water deficits are also important for reproducing vegetation dynamics (Anav et al., 2013).
**5. Conclusion**
In this study, we evaluated twenty global terrestrial ET estimates including four from remote
sensing-based physical models, two from machine-learning algorithms and fourteen from
TRENDY LSMs. The ensemble mean values of global terrestrial ET for the three categories agreed
well, ranging from 589.6 mm yr$^{-1}$ to 617.1 mm yr$^{-1}$. All of the three categories detected an overall
increasing trend in global ET during the period 1982-2011 and suggested a positive effect of
vegetation greening on ET intensification. However, the multi-model inter-comparison indicates
that, considerable uncertainties still exist in both the temporal and spatial variations in global ET
estimates. LSMs had significant differences in the ET magnitude in tropical regions especially the
Amazon Basin, while benchmark ET products showed larger inter-model range in arid and semi-
arid regions than LSMs. Trends in LSMs ET estimates also had significant discrepancies. These
uncertainties are induced by parameterization of land processes, meteorological forcing, lack of in
situ measurements, remote sensing acquisition and scaling effects. Model developments and
observational improvements provide two parallel pathways towards improving the accuracy of
global terrestrial ET estimation.
**Code and data availability**



TRENDYv6 data are available from S.S. (s.a.sitch@exeter.ac.uk) on reasonable request. MODIS
ET data are available from http://files.ntsg.umt.edu/data/NTSG_Products/MOD16/. GLEAM ET are
available from https://www.gleam.eu/. Both Model Tree Ensemble and Random Forest ET are
available from https://www.bgc-jena.mpg.de/geodb/projects/FileDetails.php. P-LSH ET are
available from http://files.ntsg.umt.edu/data/ET_global_monthly/Global_8kmResolution/. PML-
CSIRO ET are from https://data.csiro.au/dap/landingpage?pid=csiro:17375. CRU-NCEPv8 data are
available from Nicolas Viovy on reasonable request. GIMMS LAI3gV1 data are available from R.
B. Myneni on reasonable request. GIMMS NDVI3gV1 data are available from
https://ecocast.arc.nasa.gov/data/pub/gimms/3g.v1/.

**Author contributions**

S.P. initiated this research and was responsible for the integrity of the work as a whole. N.P. carried
out the analyses. S.P., N.P., H.T. and H.S wrote the manuscript with contributions from all authors.
P.F., S.S., V.K.A., V.H., A.K.J., E.K., S.L., D.L, C.O., B.P., H.T. and S.Z. contributed to the
TRENDY results.

**Competing interests**

The authors declare that they have no conflict of interest.

**Acknowledgements**

This study has been supported partially by grants from National Science Foundation (1903722 and
1243232), AU-OUC Joint Center Program and Auburn University IGP Program. Atul K Jain was
support in part by Department of Energy (No. DE‐SC0016323) and NSF (NSF AGS 12-43071).
Vanessa Haverd acknowledges support from the Earth Systems and Climate Change Hub, funded by the





Australian Government's National Environmental Science Program. We thank all people who provided
data used in this study, in particular, the TRENDY modelling groups.

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





**Table 1.** Descriptions of models used in this study, including their drivers, adopted algorithms,
key equations, limitations and references

| Name | Input | Algorithm | Spatial resolution | Temporal resolution | Key equations | Limitations | References |
|------|-------|-----------|--------------------|--------------------|----------------|-------------|------------|
| MTE | Climate: precipitation, temperature, sunshine hour, relative humidity, wet days Vegetation: fAPAR | TRIAL + ERROR | 0.5°×0.5° | Monthly | No specific equation | Insufficient flux observations in tropical regions; with no CO2 effect | Jung et al. (2011) |
| RF | enhanced vegetation index, fAPAR, leaf area index, land surface temperature, radiation, potential radiation, index of water availability, relative humidity | Randomized decision tree | 0.5°×0.5° | Half-hourly | No specific equation | The same with MTE | Bodesheim et al. (2018) |
| P-LSH | Climate: radiation, air temperature, vapor pressure, wind speed, CO2 Vegetation: AVHRR NDVI | Modified Penman–Monteith | 0.083°×0.083° | Monthly | $E_v = \dfrac{\Delta R_n + \rho C_p VPD g_a}{\lambda_v \left(\Delta + \Upsilon \left(1 + \frac{g_a}{g_s}\right)\right)}$ $E_s = RH^{\frac{VPD}{k}} \dfrac{\Delta R_n + \rho C_p VPD g_a}{\lambda_v \left(\Delta + \Upsilon \left(1 + \frac{g_a}{g_s}\right)\right)}$ | Advantages: more robust physical basis; consider the effects of CO2 Limitations: high meteorological forcingrequirements; canopy conductance is based on proxies; | Zhang et al. (2015) |
| GLEAM | Climate: precipitation, net radiation, surface soil moisture, land surface temperature, air temperature, snow depth Vegetation: vegetation optical depth | Modified Priestley–Taylor | 0.25°×0.25° | Daily | $E_s = f_s S_s \alpha_s \dfrac{\Delta}{\lambda_v \rho_w (\Delta + \gamma)} (R_n^s - G_s)$ $E_{sc} = f_{sc} S_{sc} \alpha_{sc} \dfrac{\Delta}{\lambda_v \rho_w (\Delta + \gamma)} (R_n^{sc} - G_{sc})$ $E_{tc} = f_{tc} S_{tc} \alpha_{tc} \dfrac{\Delta}{\lambda_v \rho_w (\Delta + \gamma)} (R_n^{tc} - G_{tc}) - \beta E_i$ | Advantages: simple;low requirement for meteorological data; well-suited for remote sensing observable variables; soil moisture is considered Limitations: many simplifications of physicalprocesses; neither VPD nor surface and aerodynamic resistances are explicitly accounted for; strong dependency on net radiationn | (Miralles et al., 2011) |





| Name | Algorithm | Spatial resolution | Temporal resolution | Key equations | References |
|---|---|---|---|---|---|
| MODIS | Climate: air temperature, shortwave radiation, wind speed, relative humidity, air pressure Vegetation: LAI, fAPAR, albedo | Penman–Monteith–Leuning | 0.05°×0.05° | Monthly | $E_i = f_{wet} f_c \dfrac{\Delta(R_n - G) + \rho c_p \frac{VPD}{r_a^{wc}}}{\lambda_v \rho_w (\Delta + \gamma \frac{r_s^{wc}}{r_a^{wc}})}$ $E_v = (1 - f_{wet}) f_c \dfrac{\Delta(R_n - G) + \rho c_p \frac{VPD}{r_a^{t}}}{\lambda_v \rho_w (\Delta + \Upsilon \frac{r_s^{t}}{r_a^{t}})}$ $E_s = [f_{wet} + \dfrac{(1 - f_{wet}) h VPD}{\beta}] \dfrac{(s A_{soil} + \frac{\rho c_p (1 - f_c) VPD}{r_{as}})}{\lambda_v \rho_w (S + \gamma \frac{r_{tot}}{r_{as}})}$ | Advantages: more robust physical basis; Limitations: require many variables that are difficult to observe or not observable with satellites; canopy conductance is based on proxies; do not consider soil moisture but use atmospheric humidity as a surrogate; do not consider the effects of $CO_2$ | Mu et al. (2011) |
| PML-CSIRO | Climate: precipitation, air temperature, vapor pressure, shortwave radiation, longwave radiation, wind speed Vegetation: AVHRR LAI, emissivity and albedo | Penman–Monteith–Leuning | 0.5°×0.5° | Monthly | $E_v = \dfrac{\Delta R_n + \rho C_p VPD g_a}{\lambda_v (\Delta + \Upsilon(1 + \frac{g_a}{g_s}))}$ $E_s = \dfrac{f \Delta A_s}{\Delta + \gamma}$ $E_i$: an adapted version of Gash rainfall interception model (Van et al., 2001) | Advantages: more robust physical basis (compared to Priestley–Taylor equation); biophysically based estimation of surface conductance Limitations: high meteorological forcingrequiremen ts; canopy conductance is based on proxies; do not consider the effects of $CO_2$ | Zhang et al. (2016b) |

**TRENDY LSMs**

Advantages: land surface models are process-oriented and physically-based. Given their structure almost all models are capable to allow factorial analysis, where one forcing can be applied at a time. Most models also consider the physiological effect of CO2 on stomatal closure.

Disadvantages: most models typically do not allow integration/assimilation of observation-based vegetation characteristics. Model parameterizations remain uncertain and a same process is modelled in different ways across models. Model parameters may or may not be physically-based and therefore measurable in field.

Models participating in the TRENDY 2017 comparison were forced by precipitation, air temperature, specific humidity, shortwave radiation, longwave radiation, wind speed based on the CRU-NCEPv8 data as explained in Le Quere et al. 2018. It is very difficult to list all key equations for all land surface models. Here, we just list the stomatal conductance equation for each model.

| Name | Algorithm | Spatial resolution | Temporal resolution | Key equations | References |
|---|---|---|---|---|---|
| CABLE | Penman-Monteith | 0.5°×0.5° | Monthly | $g_s = g_0 + \dfrac{g_1 f_w A}{c_a - c_p}(1 + \dfrac{VPD}{VPD_0})^{-1}$ | Haverd et al. (2018) |
| CLASS-CTEM | Modified Penman–Monteith | 2.8125°×2.8125° | Monthly | $g_c = m \dfrac{A_n p}{(c_s - \Gamma)} \dfrac{1}{(1 + VPD/VPD_0)} + b\, LAI$ | Melton and Arora (2016) |
| CLM45 | Modified Penman–Monteith | 1.875°×2.5° | Monthly | $g_s = g_0 + \dfrac{g_1 A}{c_a} RH$ | Oleson et al. (2010) |
| DLEM | Penman–Monteith | 0.5°×0.5° | Monthly | $g_s = \max(g_{smax} r_{corr} bf(ppdf) f(T_{min}) f(VPD) f(CO_2), g_{smin})$ | Pan et al. (2015) |



| ISAM | Modified Penman–Monteith | 0.5°×0.5° | Monthly | $g_s = m \dfrac{A}{C_s/_{P_{atm}}} \times \dfrac{e_s}{e_i} + b_t\beta_t$ | Barman et al. (2014) |
|---|---|---|---|---|---|
| JSBACH | Penman–Monteith | 3.913°×3.913° | Monthly | $g_s = \beta_w \dfrac{1.6A_{n,pot}}{c_a - c_{i,pot}}$ | Knauer et al. (2015) |
| JULES | Penman–Monteith | 2.5°×3.75° | Monthly | Bare soil conductance: $g_{soil} = \dfrac{1}{100}\left(\dfrac{\theta_1}{\theta_c}\right)^2$ <br> Stomatal conductance is calculated by solving the two equations: <br> $A_l = g_s(C_s - C_i)/1.6$; <br> $\dfrac{C_i - \Gamma^*}{C_c - \Gamma^*} = f_0\left(1 - \dfrac{\Delta}{q_c}\right)$ | Li et al. (2016) |
| LPJ-GUESS | Equations proposed by Monteith (1995) | 0.5°×0.5° | Monthly | $g_s = g_{smin} + \dfrac{1.6A_{dt}}{c_a(1 - \lambda_c)}$ | Smith (2001) |
| LPJ-wsl | Priestley-Taylor | 0.5°×0.5° | Monthly | $g_s = g_{smin} + \dfrac{1.6A_{dt}}{c_a(1 - \lambda_c)}$ | Sitch et al. (2003) |
| LPX-Bern | Modified equation of Monteith (1995) | 1°×1° | Monthly | $g_s = g_{smin} + \dfrac{1.6A_{dt}}{c_a(1 - \lambda_c)}$ | Keller et al. (2017) |
| O-CN | Modified Penman-Monteith | 1°×1° | Monthly | $g_s = g_{smin} + \dfrac{1.6A_{dt}}{c_a(1 - \lambda_c)}$ | Zaehle and Friend (2010) |
| ORCHIDEE | Modified Penman-Monteith | 0.5°×0.5° | Monthly | $g_s = g_0 + \dfrac{A + R_d}{c_a - c_p}f_{vpd}$ <br><br> $g_{soil} = \exp(8.206 - 4.255 W/W_{sat})$ | d'Orgeval et al. (2008) |
| ORCHIDEE-MICT | Modified Penman-Monteith | 0.5°×0.5° | Monthly | $g_s = g_0 + \dfrac{A + R_d}{c_a - c_p}f_{vpd}$ | Guimberteau et al. (2018) |
| VISIT | Penman–Monteith | 0.5°×0.5° | Monthly | $g_s = g_0 + \dfrac{g_1 f_w A}{c_a - c_p}\left(1 + \dfrac{VPD}{VPD_0}\right)^{-1}$ | Ito (2010) |

Notes: A: net assimilation rate; $A_{dt}$: total daytime net photosynthesis; $A_{n,pot}$: unstressed net
assimilation rate; b: soil moisture factor; $b_t$: stomatal conductance intercept; $c_a$: atmospheric $CO_2$
concentration; $c_c$: critical $CO_2$ concentration; $c_i$: internal leaf concentration of $CO_2$; $c_{i,pot}$: internal
leaf concentration of $CO_2$ for unstressed conditions; $c_s$: leaf surface $CO_2$ concentration; $c_p$: $CO_2$
compensation point; $e_s$: vapor pressure at leaf surface; $e_i$: saturation vapor pressure inside the leaf;
$E_s$: soil evaporation; $E_c$: canopy evapotranspiration; $E_{dry}$: dry canopy evapotranspiration; $E_{wet}$: wet
canopy evapotranspiration; $E_v$: canopy transpiration; $E_i$: canopy interception; $E_{tc}$: transpiration
from tall canopy; $E_{sc}$: transpiration from short canopy; f: fraction of *P* to equilibrium soil
evaporation; $f_s$: soil fraction; $f_{sc}$: short canopy fraction; $f_{tc}$: tall canopy fraction; $f_{vpd}$: factor of the
effect of leaf-to-air vapor pressure difference; $f_w$: a function describing the soil water stress on
stomatal conductance; $f_{wet}$: relative surface wetness parameter; $f_0$: the maximum ratio of internal
to external $CO_2$; *f(ppdf)*: limiting factor of photosynthetic photo flux density; *f($T_{min}$)*: limiting factor
of daily minimum temperature; *f(VPD)*: limiting factor of vapor pressure deficit; *f($CO_2$)*: limiting
factor of carbon dioxide; G: ground energy flux; $g_a$: aerodynamic conductance; $g_m$:
empiricalparameter; $g_s$: stomatal conductance; $g_{smax}$: maximum stomatal conductance; $g_{smin}$:
minimum stomatal conductance; $g_{soil}$: bare soil conductance; $g_0$: residual stomatal conductance
when the net assimilation rate is 0 ; $g_1$: sensitivity of stomatal conductance to assimilation, ambient
$CO_2$ concentration and environmental controls; I: tall canopy interception loss; m: stomatal
conductance slope; $P_{atm}$: atmospheric pressure; $PE_s$: potential soil evaporation; $PE_{canopy}$: potential
canopy evaporation; $q_a$: specific air humidity; $q_c$: critical humidity deficit; $q_s$: specific humidity of
saturated air; $r_a$: aerodynamic resistance; $r_s$: stomatal resistance; $R_n$: net radiation; Rd: day





respiration; RH: relative humidity; $T_s$: actual surface temperature; VPD: vapor pressure deficit; $VPD_0$: the sensitivity of stomatal conductance to VPD; W: top soil moisture; $W_{canopy}$: canopy water; $W_{sat}$: soil porosity; α: Priestley-Taylor coefficient; $α_m$: empirical parameter; $β$: a constant accounting for the times in which vegetation is wet; $β_t$: soil water availability factor between 0 and 1; $β_w$: empirical water stress factor which is a linear function of soil water content; $β_s$: moisture availability function; ρ: air density; γ: psychrometric constant; $λ_v$: latent heat of vaporization; $λ_c$: ratio of intercellular to ambient partial pressure of $CO_2$; $r_{corr}$: correction factor of temperature and air pressure on conductance; $Γ^*$: $CO_2$ compensation point when leaf day respiration is zero; $θ_1$: parameter of moisture concentration in the top soil layer; $θ_c$: parameter of moisture concentration in the spatially varying critical soil moisture; Δ: slope of the vapor pressure curve.





1136    **Table 2**. Inter-annual variability (IAV, denoted as standard deviation) and trend of global

1137    terrestrial ET during 1982-2011 and the contribution of vegetation greening to ET trend. * suggests

1138    significance of the trend at the 95% confidence level ($p<0.05$).

| | Model | ET Trend (mm yr$^{-2}$) | Greening-induced ET change (mm yr$^{-2}$) | Sensitivity of ET to LAI (mm yr$^{-2}$ per m$^2$ m$^{-2}$) | LAI trend (10$^{-3}$ m$^2$ m$^{-2}$ yr$^{-1}$) |
|---|---|---|---|---|---|
| Machine learning | MTE | 0.38* | 0.09 | 35.86 | 2.51* |
| RS models | P-LSH | 1.07* | 0.34 | 135.46 | 2.51* |
| | GLEAM | 0.33* | 0.14 | 55.78 | 2.51* |
| | PML-CSIRO | 0.41* | 0.36 | 143.43 | 2.51* |
| | RS model mean | 0.62* | 0.28 | 111.55 | 2.51* |
| LSMs | CABLE | 0.07 | 0.35 | 102.64 | 3.41* |
| | CLASS-CTEM | 0.35* | 0.53 | 134.52 | 3.94* |
| | CLM45 | 0.38* | 0.31 | 67.54 | 4.59* |
| | DLEM | 0.26* | 0.53 | 200.76 | 2.64* |
| | ISAM | 0.22 | 0.16 | 32.26 | 4.96* |
| | JSBACH | -0.05 | 0.50 | 217.39 | 2.30* |
| | JULES | -0.02 | 0.34 | 85.21 | 3.99* |
| | LPJ-GUESS | 0.50* | 0.28 | 160.92 | 1.74* |
| | LPJ-wsl | 0.24* | 0.19 | 31.56 | 6.02* |
| | LXP-Bern | 0.20* | 0.04 | 4.04 | 9.90* |
| | O-CN | 0.32* | 0.53 | 89.23 | 5.94* |
| | ORCHIDEE | -0.17 | 0.21 | 96.33 | 2.18* |
| | ORCHIDEE-MICT | -0.34* | 0.50 | 171.23 | 2.92* |
| | VISIT | 0.87* | 0.70 | 51.40 | 13.62* |
| | LSM mean | 0.23 | 0.37 | 79.91 | 4.63* |