# Peer review of "Evaluation of global terrestrial evapotranspiration by state-of-the-art"

_Hydrology and Earth System Sciences, 2019_

## Referee Comment (RC1) · Anonymous Referee #1 · 10 Sep 2019

The authors provide a nice refresh reviewing global ET data products. Generally, it's a good literature review.

Overall, however, the paper is excessively long and unfocused. Basically, the authors took a bunch of data products, calculated different comparative statistics, and discussed some patterns. That said, the title accurately depicts the unfocused nature of the paper, so it should not come as a surprise. The authors did try to throw in some science by looking at controls over ET, but this only served to make the paper even longer and more spread thin. Moreover, this type of product review has already been done by Mueller, Jimenez and others, so the novelty here is light. The science focus

and strength are mostly on the land surface models, while the remote sensing is noticeably weak (there might be zero ET remote sensing authors on the list of 15 authors). The balanced title does not reflect the unbalanced paper.

In general, I liked the paper as a source for a lit review.

Some additional references that may be useful:

• Talsma, C.J., 2018. Sensitivity of evapotranspiration components in remote sensing-based models. Remote Sensing 10(1601): 1-28. • Talsma, C., 2018. Partitioning of evapotranspiration in remote sensing-based models. Agricultural and Forest Meteorology 260-261: 131-143. • Jiménez, C., 2018. Exploring the merging of the global land evaporation WACMOS-ET product based on local measurements. Hydrology and Earth System Sciences 22(8): 4513-4533. • Badgley, G., 2015. On uncertainty in global terrestrial evapotranspiration estimates from choice of input forcing datasets. Journal of Hydrometeorology 16(4): 1449-1455. • Polhamus, A.M., 2012. What controls the error structure in evapotranspiration models? Agricultural and Forest Meteorology 169: 12-24. • Fisher, J.B., 2011. ET Come Home: Potential evapotranspiration in geographical ecology. Global Ecology and Biogeography 20: 1-18.

---

## Referee Comment (RC2) · Anonymous Referee #2 · 12 Sep 2019

This paper was already well-written, especially for the detailed discussion of limitations and possible next steps of different ET products. The reviewer thus has a few minor suggestions for the authors to consider. 1. The remote-sensing based, machine learning, and LSMs ET were comprehensively intercompared. However, how is the performance of ET outputs from the Earth system models (e.g., those from CMIP5 and CMIP6) and the reanalysis? There must be a reason why the authors did not include them. But please clarify this or add these comparison results. 2. In lines 245-246, you indicated the benchmarking products are from the machine learning and physical-based satellite datasets. It seems confusing both here and in Figs 3, 5, and 7. For example, in Fig. 7, if the benchmark product is the simple combination of the two data

sources, the variation of benchmark product (yellow line), which is the averaged value, should be largely in the middle of the remote sensing (green line) and machine learning (blue line) results. Please double check this. 3. The Abstract seems quite long. Please double check if the Abstract length fits this particular journal. 4. In line 483, Fig. 5 does not have subfigures.

—————————————————————

---

## Author Comment (AC1) · 10 Nov 2019

To Reviewer #1: General Response: We appreciate the reviewer for the positive comments. We have addressed all your comments and cited the references you recommended. Below are the reviewer's comments, followed by our responses and changes in manuscript.

\*\*\*\*\*\*\*\*\*\*\* [Reviewer #1 General Comment] The authors provide a nice refresh reviewing global ET data products. Generally, it's a good literature review. Overall, however, the paper is excessively long and unfocused. Basically, the authors took a bunch of data products, calculated different comparative statistics, and discussed some patterns. That said, the title accurately depicts the unfocused nature of the paper, so it should not come as a surprise. The authors did try to throw in some science by looking at controls over ET, but this only served to make the paper even longer and more spread thin. Moreover, this type of product review has already been done by Mueller, Jimenez and others, so the novelty here is light. The science focus and strength are mostly on the land surface models, while the remote sensing is noticeably weak (there might be zero ET remote sensing authors on the list of 15 authors). The balanced title does not reflect the unbalanced paper. In general, I liked the paper as a source for a lit review. [Response] We thank the reviewer for the positive comments. We admit that our paper is long. It is mainly because our study included a plenty of ET products of different types and we reviewed their principles, advantages, disadvantages and future directions. However, we think these descriptions and discussions are necessary because they give readers a comprehensive understanding in the strengths and limitations of each ET model and shows them possible solutions for overcoming the uncertainties identified in our analyses. As you stated, Mueller et al. (2011) and Jimenez et al. (2011) conducted analyses on different ET products. Nevertheless, the focus of our paper is different from theirs. Mueller et al. (2011) mainly focused on comparing IPCC AR4 ET estimates and observations-based ET estimates. Jimenez et al. (2011) mainly focused on the intercomparison of the seasonal variability of different latent heat, sensible heat and net radiative heat fluxes. Little discussion on the source of uncertainty and suggestions for future development was given in their papers. In comparison, our study emphasized on the analyses of uncertainty sources in different types of ET estimations and on the solutions for overcoming these identified uncertainties. In addition, our study incorporated ET estimates from fourteen state-of-the-art land surface models joining in the Trends and Drivers of the Regional Scale Sources and Sinks of Carbon Dioxide (TRENDY) Project, which is our strength over the previous studies. We want to clarify that although there is no ET remote sensing authors on the list of 15 authors, the parts regarding remote sensing-based physical models have similar length with that of land surface models and machine learning algorithms

in the text. As a synthesis of ET estimates from different approaches, we didn't focus too much on either land surface models or remote sensing-based models. According to the references you recommended, we added citations and several sentences about the future development of remote sensing based ET models (in Section 4.2.1). "Most existing remote sensing-based ET studies focused on total ET, however, the partitioning of ET between transpiration, soil evaporation, and canopy interception may have significant divergence even though the total ET is accurately estimated (Talsma et al., 2018). In current remote sensing-based ET models, soil evaporation which is sensitive to precipitation events and soil moisture is the part with the largest error, therefore incorporating the increasing accessible satellite-based precipitation, soil moisture observations and soil property data will contribute to the improvement of soil evaporation estimation. Meanwhile, the consideration of soil evaporation under herbaceous vegetation and canopy will also reduce the errors." References Jimenez, C., Prigent, C., Mueller, B., Seneviratne, S.I., McCabe, M., Wood, E., Rossow, W., Balsamo, G., Betts, A., Dirmeyer, P. (2011) Global intercomparison of 12 land surface heat flux estimates. Journal of Geophysical Research: Atmospheres 116. Mueller, B., Seneviratne, S.I., Jimenez, C., Corti, T., Hirschi, M., Balsamo, G., Ciais, P., Dirmeyer, P., Fisher, J., Guo, Z. (2011) Evaluation of global observations‐based evapotranspiration datasets and IPCC AR4 simulations. Geophysical Research Letters 38. Talsma, C.J., Good, S.P., Jimenez, C., Martens, B., Fisher, J.B., Miralles, D.G., McCabe, M.F., Purdy, A.J. (2018) Partitioning of evapotranspiration in remote sensing-based models. Agricultural and Forest Meteorology 260-261, 131-143.

---

## Author Comment (AC2) · 10 Nov 2019

To Reviewer #2:

General Response: We appreciate the reviewer for the positive comments. We have addressed the stated comments point-by-point. Below are the reviewer's comments, followed by our responses and changes in manuscript.

\*\*\*\*\*\*\*\*\*\*\*

[Reviewer #2 General Comment] This paper was already well-written, especially for the detailed discussion of limitations and possible next steps of different ET products. The

reviewer thus has a few minor suggestions for the authors to consider. [Response] We appreciate the reviewer for the positive comments.

[Reviewer #2 Specific Comment 1] The remote-sensing based, machine learning, and LSMs ET were comprehensively intercompared. However, how is the performance of ET outputs from the Earth system models (e.g., those from CMIP5 and CMIP6) and the reanalysis? There must be a reason why the authors did not include them. But please clarify this or add these comparison results. [Response] Thanks for pointing out this issue. We didn't include ET outputs from the Earth system models (e.g., those from CMIP5 and CMIP6) because previous study confirmed systematic biases in global terrestrial ET estimated by CMIP5 models (Mueller and Seneviratne, 2014) and CMIP6 data were not available when we conducted our analyses. Reanalysis systems which are built upon the assimilation of extensive disparate observations in a physically consistent manner are capable of providing the estimates for a broad range of variables (Balsamo et al., 2015; Rienecker et al., 2011). ET estimates derived from both atmospheric and off-line land reanalysis datasets have been evaluated at local, regional and global scales (Baik et al., 2018; Feng et al., 2019; Mao and Wang, 2017) and have been compared with estimates from other approaches (Jimenez et al., 2011; Mueller et al., 2013; Mueller et al., 2011). The objective of this study is to identify the uncertainty sources in each type of ET estimations. However, these reanalysis systems integrate multiple process modules, multi-source remote sensing observations and ground-based measurements, and multiple assimilation algorithms, which lead to the mixture of systematic errors and make it hard to identify the sources of errors in ET estimations at the global scale. For above-mentioned reasons, our analyses didn't include ET outputs from the Earth system models and the reanalysis.

[Reviewer #2 Specific Comment 2] In lines 245-246, you indicated the benchmarking products are from the machine learning and physical-based satellite datasets. It seems confusing both here and in Figs 3, 5, and 7. For example, in Fig. 7, if the benchmark product is the simple combination of the two data [Response] The ensemble mean of

benchmark products was calculated as the mean value of all machine learning and physical-based satellite estimates (6 datasets for Fig. 3 and 5, and 5 datasets for Fig. 7) rather than the mean value of machine learning ensemble mean and satellite ensemble mean, since we treated each benchmark dataset equally. We have added the sentence describing the calculation of the ensemble mean of benchmark products in Section 2.2 The ensemble mean of benchmark products was calculated as the mean value of all machine learning and physical-based satellite estimates, since we treated each benchmark dataset equally.

[Reviewer #2 Specific Comment 3] The Abstract seems quite long. Please double check if the Abstract length fits this particular journal. [Response] We have double checked journal's requirements for manuscript, there is no particular limitation on the length of abstract. Following your comment, we have shortened the abstract. Evapotranspiration (ET) is a critical component in global water cycle and links terrestrial water, carbon and energy cycles. Accurate estimate of terrestrial ET is important for hydrological, meteorological, and agricultural research and applications. However, direct measurement of global terrestrial ET is not feasible. Here, we first gave a retrospective introduction to the basic theory and recent developments of state-of-the-art approaches for estimating global terrestrial ET, including remote sensing-based physical models, machine learning algorithms and land surface models (LSMs). Then, we utilized six remote sensing-based models (including four physical models and two machine learning algorithms) and fourteen LSMs to analyze the spatial and temporal variations in global terrestrial ET. The results showed that the mean annual global terrestrial ET ranged from 50.7×103 km3 yr-1 (454 mm yr-1) to 75.7 ×103 km3 yr-1 (697 mm yr-1), with the average being 65.5×103 km3 yr-1(588 mm yr-1). LSMs had significant uncertainty in the ET magnitude in tropical regions especially the Amazon Basin, while remote sensing-based ET products showed larger inter-model range in arid and semi-arid regions than LSMs. LSMs and remote sensing-based physical models presented much larger inter-annual variability (IAV) of ET than machine learning algorithms in southwestern U.S. and the Southern Hemisphere, particularly in Australia. LSMs sug-

gested stronger control of precipitation on ET IAV than remote sensing-based models. During 1982-2011, the ensemble remote sensing-based physical models and machine-learning algorithm suggested significant increasing trends in global terrestrial ET at the rate of 0.62 mm yr-2 (p<0.05) and 0.38 mm yr-2 (p<0.05), respectively. In contrast, the ensemble mean of LSMs showed no statistically significant change (0.23 mm yr-2, p>0.05), even though most of the individual LSMs reproduced the increasing trend. Moreover, all models suggested a positive effect of vegetation greening on ET intensification. In general, the ensemble means of the three ET categories showed generally good consistency, however, considerable uncertainties still exist in both the temporal and spatial variations in global ET estimates. The uncertainties were induced by multiple factors, including parameterization of land processes, meteorological forcing, lack of in situ measurements, remote sensing acquisition and scaling effects. Improvements in the representation of water stress and canopy dynamics are essentially needed to reduce uncertainty in LSM-simulated ET. Utilization of latest satellite sensors and deep learning methods, theoretical advancements in non-equilibrium thermodynamics, and application of integrated methods that fuse different ET estimates or relevant key biophysical variables will improve the accuracy of remote sensing-based models.

[Reviewer #2 Specific Comment 4] In line 483, Fig. 5 does not have subfigures. [Response] We are sorry for the wrong numbering. "Fig. 5c-d" in our previous manuscript should be "Fig. 8c-d". We have corrected this error in the main text.

References Baik, J., Liaqat, U.W., Choi, M. (2018) Assessment of satellite- and reanalysis-based evapotranspiration products with two blending approaches over the complex landscapes and climates of Australia. Agricultural and Forest Meteorology 263, 388-398. Balsamo, G., Albergel, C., Beljaars, A., Boussetta, S., Brun, E., Cloke, H., Dee, D., Dutra, E., Muñoz-Sabater, J., Pappenberger, F. (2015) ERA-Interim/Land: a global land surface reanalysis data set. Hydrology and Earth System Sciences 19, 389-407. Feng, T., Su, T., Zhi, R., Tu, G., Ji, F. (2019) Assessment of actual evapotranspiration variability over global land derived from seven reanalysis datasets. International Journal of Climatology 39, 2919-2932. Mao, Y., Wang, K. (2017) Comparison of evapotranspiration estimates based on the surface water balance, modified Penman‐Monteith model, and reanalysis data sets for continental China. Journal of Geophysical Research: Atmospheres 122, 3228-3244. Mueller, B., Hirschi, M., Jimenez, C., Ciais, P., Dirmeyer, P., Dolman, A., Fisher, J., Jung, M., Ludwig, F., Maignan, F. (2013) Benchmark products for land evapotranspiration: LandFlux-EVAL multi-data set synthesis. Hydrology and Earth System Sciences. Mueller, B., Seneviratne, S.I. (2014) Systematic land climate and evapotranspiration biases in CMIP5 simulations. Geophysical Research Letters 41, 128-134. Rienecker, M.M., Suarez, M.J., Gelaro, R., Todling, R., Bacmeister, J., Liu, E., Bosilovich, M.G., Schubert, S.D., Takacs, L., Kim, G.-K., Bloom, S., Chen, J., Collins, D., Conaty, A., da Silva, A., Gu, W., Joiner, J., Koster, R.D., Lucchesi, R., Molod, A., Owens, T., Pawson, S., Pegion, P., Redder, C.R., Reichle, R., Robertson, F.R., Ruddick, A.G., Sienkiewicz, M., Woollen, J. (2011) MERRA: NASA's Modern-Era Retrospective Analysis for Research and Applications. Journal of Climate 24, 3624-3648.

———————————————————————

---

## Author Response (AR1)

**To Reviewer #1:**

We appreciate the reviewer for the positive comments. We have responded to all your comments and cited the references you recommended. Below are the reviewer's comments, followed by our responses and changes in manuscript.

Sincerely,

Shufen Pan (on behalf of the author team)

\*\*\*\*\*\*\*\*\*\*

**[Reviewer #1 General Comment]** The authors provide a nice refresh reviewing global ET data products. Generally, it's a good literature review. Overall, however, the paper is excessively long and unfocused. Basically, the authors took a bunch of data products, calculated different comparative statistics, and discussed some patterns. That said, the title accurately depicts the unfocused nature of the paper, so it should not come as a surprise. The authors did try to throw in some science by looking at controls over ET, but this only served to make the paper even longer and more spread thin. Moreover, this type of product review has already been done by Mueller, Jimenez and others, so the novelty here is light. The science focus and strength are mostly on the land surface models, while the remote sensing is noticeably weak (there might be zero ET remote sensing authors on the list of 15 authors). The balanced title does not reflect the unbalanced paper. In general, I liked the paper as a source for a lit review.

**[Response]** We thank the reviewer for the positive comments. We admit that our paper is long. It is mainly because our study included a plenty of ET products of different types and we reviewed their principles, advantages, disadvantages and future directions. However, we think these descriptions and discussions are necessary because they give readers a comprehensive understanding in the strengths and limitations of each ET model and shows them possible solutions for overcoming the uncertainties identified in our analyses. As you stated, Mueller et al. (2011) and Jimenez et al. (2011) conducted analyses on different ET products. Nevertheless, the focus of our paper is different from theirs. Mueller et al. (2011) mainly focused on comparing IPCC AR4

ET estimates and observations-based ET estimates. Jimenez et al. (2011) mainly focused on the intercomparison of the seasonal variability of different latent heat, sensible heat and net radiative heat fluxes. Few discussion on the source of uncertainty and suggestions for future development was given. In comparison, our study emphasized on the analyses of uncertainty sources in different types of ET estimations and on the solutions for overcoming these identified uncertainties. In addition, our study incorporated ET estimates from fourteen state-of-the-art land surface models joining in the Trends and Drivers of the Regional Scale Sources and Sinks of Carbon Dioxide (TRENDY) Project, which is our strength over the previous studies. We want to clarify that although there is no ET remote sensing author on the list of 15 authors of our first version, the parts regarding remote sensing-based physical models have similar length with that of land surface models and machine learning algorithms in the text. As a synthesis of ET estimates from different approaches, we didn't focus too much on either land surface models or remote sensing-based models. In addition, Steven W Running, an expert in the area of remote sensing based ET, joined our author team and proposed several constructive suggestions which improved our manuscript. We proposed that terrestrial ET also has a potential planetary boundary (Page32 Line617-629 of the revised manuscript).

According to the references you recommended, we added citations and several sentences about the future development of remote sensing based ET models (Page35 Line671-679 of the revised manuscript).

"Most existing remote sensing-based ET studies focused on total ET, however, the partitioning of ET between transpiration, soil evaporation, and canopy interception may have significant divergence even though the total ET is accurately estimated (Talsma et al., 2018b). In current remote sensing-based ET models, soil evaporation, which is sensitive to precipitation events and soil moisture, is the part with the largest error (Talsma et al., 2018a). Therefore incorporating the increasing accessible satellite-based precipitation, soil moisture observations and soil property data will contribute to the improvement of soil evaporation estimation. Meanwhile, the consideration of soil evaporation under herbaceous vegetation and canopy will also reduce the errors."

**To Reviewer #2:**

We appreciate the reviewer for the positive comments. We have addressed the stated comments point-by-point. Below are the reviewer's comments, followed by our responses and changes in manuscript.

Sincerely,

Shufen Pan (on behalf of the author team)

\*\*\*\*\*\*\*\*\*\*\*

**[Reviewer #2 General Comment]** This paper was already well-written, especially for the detailed discussion of limitations and possible next steps of different ET products. The reviewer thus has a few minor suggestions for the authors to consider.

**[Response]** We appreciate the reviewer for the positive comments.

**[Reviewer #2 Specific Comment 1]** The remote-sensing based, machine learning, and LSMs ET were comprehensively intercompared. However, how is the performance of ET outputs from the Earth system models (e.g., those from CMIP5 and CMIP6) and the reanalysis? There must be a reason why the authors did not include them. But please clarify this or add these comparison results.

**[Response]** Thanks for pointing out this issue. We didn't include ET outputs from the Earth system models (e.g., those from CMIP5 and CMIP6) because previous study confirmed systematic biases in global terrestrial ET estimated by CMPI5 models (Mueller and Seneviratne, 2014) and CMIP6 data were not available when we conducted our analyses. Reanalysis systems which are built upon the assimilation of extensive disparate observations in a physically consistent manner are capable of providing the estimates for a broad range of variables (Balsamo et al., 2015; Rienecker et al., 2011). ET estimates derived from both atmospheric and off-line land reanalysis datasets have been evaluated at local, regional and global scales (Baik et al., 2018; Feng et al., 2019; Mao and Wang,

2017) and have been compared with estimates from other approaches (Jimenez et al., 2011; Mueller et al., 2013; Mueller et al., 2011). The objective of this study is to identify the uncertainty sources in each type of ET estimations. However, these reanalysis systems integrate multiple process modules, multi-source remote sensing observations and ground-based measurements, and multiple assimilation algorithms, which lead to the accumulation of systematic errors and makes it hard to identify the sources of errors in ET estimations at the global scale. For above-mentioned reasons, our analyses didn't include ET outputs from the Earth system models and the reanalysis.

**[Reviewer #2 Specific Comment 2]** In lines 245-246, you indicated the benchmarking products are from the machine learning and physicalbased satellite datasets. It seems confusing both here and in Figs 3, 5, and 7. For example, in Fig. 7, if the benchmark product is the simple combination of the two data

**[Response]** The ensemble mean of benchmark products was calculated as the mean value of all machine learning and physical-based satellite estimates (6 datasets for Fig. 3 and 5, and 5 datasets for Fig. 7) rather than the mean value of machine learning ensemble mean and satellite ensemble mean, since we treated each benchmark dataset equally. We have added the sentence describing the calculation of the ensemble mean of benchmark products in section 2.2 (Page11 Line243-245).

"The ensemble mean of benchmark products was calculated as the mean value of all machine learning and physical-based satellite estimates since we treated each benchmark dataset equally."

**[Reviewer #2 Specific Comment 3]** The Abstract seems quite long. Please double check if the Abstract length fits this particular journal.

**[Response]** We have double checked journal's requirements for manuscript, there is no particular limitation on the length of abstract. Following your comment, we have shortened the abstract.

"Evapotranspiration (ET) is critical in linking global water, carbon and energy cycles. Yet direct measurement of global terrestrial ET is not feasible. Here, we first summarized the basic theory and state-of-the-art approaches for estimating global terrestrial ET, including remote sensing-based physical models, machine learning algorithms and land surface models (LSMs). We then utilized four remote sensing-based physical models, two machine-learning algorithms and fourteen LSMs to analyze the spatial and temporal variations in global terrestrial ET. The results showed that the ensemble means of annual global terrestrial ET estimated by these three categories of approaches agreed well, ranging from 589.6 mm yr$^{-1}$ to 617.1 mm yr$^{-1}$. For the period 1982-2011, both the ensembles of remote sensing-based physical models and machine-learning algorithms suggested positive trends in global terrestrial ET (0.62 mm yr$^{-2}$, $p<0.05$ and 0.38 mm yr$^{-2}$, $p<0.05$, respectively). In contrast, the ensemble mean of LSMs showed no statistically significant change (0.23 mm yr$^{-2}$, $p>0.05$), even though many of the individual LSMs reproduced a positive trend. Nevertheless, all the twenty models used in this study showed anthropogenic earth greening had a positive role in increasing terrestrial ET. The concurrent small inter-annual variability, i.e. relative stability, found in all estimates of global terrestrial ET, suggests there exists a potential planetary boundary in regulating global terrestrial ET, with the value being about 6.74×10$^4$ km$^3$ yr$^{-1}$ (603 mm yr$^{-1}$). Uncertainties among approaches were identified in specific regions, particularly in the Amazon Basin and arid/semi-arid regions. Improvements in parameterizing water stress and canopy dynamics, utilization of new available satellite retrievals and deep learning methods, and model-data fusion will advance efforts in terrestrial ET estimates."

**[Reviewer #2 Specific Comment 4]** In line 483, Fig. 5 does not have subfigures.

**[Response]** We are sorry for the wrong numbering. "Fig. 5c-d" in our previous manuscript should be "Fig. 8c-d". We have corrected this error in the main text.

[revised manuscript text omitted]